

**Summertime observations of ultrafine particles and cloud**
**condensation nuclei from the boundary layer to the free troposphere**
**in the Arctic**
Julia Burkart[1], Megan D. Willis[1], Heiko Bozem[2], Jennie L. Thomas[3], Kathy Law[3],
Peter Hoor[2], Amir A. Aliabadi[4], Franziska Köllner[5], Johannes Schneider[5], Andreas
Herber[6], Jonathan P. D. Abbatt[1], W. Richard Leaitch[7]
[1]Department of Chemistry, University of Toronto, Toronto, Canada
[2]Institute of Atmospheric Physics, Johannes Gutenberg-University, Mainz, Germany
[3]LATMOS/IPSL, UPMC Univ. Paris 06 Sorbonne Universités, UVSQ, CNRS, Paris, France
[4]Environmental Engineering Program, University of Guelph, Guelph, Canada
[5]Particle Chemistry Department, Max Planck Institute for Chemistry, Mainz, Germany
[6]Alfred Wegener Institute, Helmholtz Center for Polar and Marine Research, Bremerhaven,
Germany
[7]Environment and Climate Change Canada, Toronto, Ontario, Canada
*Correspondence to*: J. Burkart (jburkart@chem.utoronto.ca)
**Abstract.** The Arctic is extremely sensitive to climate change. Shrinking sea ice extent increases
the area covered by open ocean during Arctic summer, which impacts the surface albedo and
aerosol and cloud properties among many things. In this context extensive aerosol measurements
(aerosol composition, particle number and size, cloud condensation nuclei, and trace gases) were
made during 11 flights of the NETCARE July, 2014 airborne campaign conducted from Resolute
Bay, Nunavut (74N, 94W). Flights routinely included vertical profiles from about 60 to 3000 m
a.g.l. as well as several low-level horizontal transects over open ocean, fast ice, melt ponds, and
polynyas.
Here we discuss the vertical distribution of ultrafine particles (UFP, particle diameter, dp: 5 – 20
nm), size distributions of larger particles (dp: 20 nm to 1 μm), and cloud condensation nuclei
(CCN, supersaturation = 0.6%) in relation to meteorological conditions and underlying surfaces.



UFPs were observed predominantly within the boundary layer, where concentrations were often
several hundreds to a few thousand particles per cubic centimeter. Occasionally, particle
concentrations below 10 cm$^{-3}$ were found. The highest UFP concentrations were observed above
open ocean and at the top of low-level clouds, whereas numbers over ice-covered regions were
substantially lower. Overall, UFP formation events were frequent in a clean boundary layer with
a low condensation sink. In a few cases this ultrafine mode extended to sizes larger than 40 nm,
suggesting that these UFP can grow into a size range where they can impact clouds and therefore
climate.

## 1    Introduction

Surface temperatures within the Arctic are rising almost twice as fast as in any other region of the
world. As a manifestation of this rapid change the summer sea ice extent has been retreating
dramatically over the past decades with the possibility that the Arctic might be ice free by the end
of this century (Boé et al., 2009) or even earlier (Wang and Overland, 2012). Increased open ocean
is likely to change the properties of both aerosol particles and clouds within the Arctic. Arctic
aerosol is well known to show a distinct seasonal variation with maximum mass concentrations
and a strong long-range anthropogenic influence in winter and early spring, known as Arctic haze
(e.g. Law and Stohl, 2007; Quinn et al., 2007; Shaw, 1995). In contrast, during summer the Arctic
is more isolated from remote anthropogenic sources and represents a comparatively pristine
environment. The reason is that the Arctic front, which provides a meteorological barrier for lower-
level air mass exchange, moves north of many source regions during the summer months.
Anthropogenic and biomass burning aerosols are transported to the Arctic during the summer, but
at the same time increased aerosol scavenging helps maintain the pristine conditions near the
surface  (e.g. Browse et al., 2012; Croft et al., 2015; Garrett et al., 2011).
Zhang et al. (2010) discuss the impacts of declining sea ice on the marine planktonic ecosystem,
which includes increasing emissions of dimethyl sulfide (DMS) that may contribute to particle
formation, such as sulphate particles, in the atmosphere (e.g. Charlson et al., 1987; Pirjola et al.,
2000). Enhanced secondary organic aerosol from emissions of biogenic volatile organic
compounds is also a possibility (Fu et al., 2009). Primary emissions of aerosol particles from the





ocean, such as sea salt and marine primary organic aerosol, may also increase (Browse et al., 2014).
Open water tends to increase cloudiness, which means that aerosol influences on clouds are likely
to be more important. Over the Arctic the effects of aerosols on clouds are especially uncertain.
Models have predicted that increasing numbers of particles may lead to overall warming (Garrett,
2004) when the atmosphere exists in a particularly low particle number state now referred to being
"CCN limited" (Mauritsen et al., 2011), to an overall cooling effect when increasing numbers of
particles are added to an atmosphere with more particles already present (Lohmann and Feichter,
2005; Twomey, 1974). It is important to characterize particle size distributions in this pristine
environment to provide a baseline against which future measurements can be compared in a
warming world. Indeed, Carslaw et al. (2013) highlight the need to understand pre-industrial like
environments with only natural aerosols in order to reduce the uncertainty in estimations of the
anthropogenic aerosol radiative forcing.
Primary sources, gas-to-particle formation processes, cloud processing, atmospheric aging, mixing
and deposition are all reflected in the size distribution. Therefore, measurements of aerosol size
distributions are important for understanding the processes particles undergo in addition to their
potential effects on clouds. The presence of ultrafine particles indicates recent production as their
lifetime is on the order of hours. We focus this paper on ultrafine particles as these are an indication
for in-situ aerosol production processes in the Arctic. We also consider the growth of newly formed
particles, as that determines how important they will be for climate.
Aerosol size distributions including ultrafine particles ($dp < 20$ nm) have been measured before at
different locations throughout the Arctic. Long term studies at ground stations such as Alert
(Leaitch et al., 2013), Ny Alesund and Zeppelin (Engvall et al., 2007; Ström et al., 2003, 2009;
Tunved et al., 2013), both on Svalbard and very recently in Tiksi, Russia (Asmi et al., 2016) and
Station Nord, Greenland (Nguyen et al., 2016) indicate a strong seasonal dependence of the size
distribution with the accumulation mode aerosol dominating during the winter months and a shift
to smaller particles during the summer months. New particle formation events are frequently
observed from June to August. Ström et al. (2003) show that the size distribution undergoes a
rapid change from an accumulation mode dominated distribution (main mode number density:
>70nm) during the winter months to an Aitken mode dominated distribution (main mode number



density: 30nm) at the beginning of summer. Total number concentrations increase at the beginning
of summer and roughly follow the incoming solar radiation on a seasonal scale suggesting that
photochemistry is an important factor for new particle formation in the Arctic. At Ny Alesund
maximum number concentrations occur in late summer and are explained by the Siberian tundra
being a potential source of aerosol precursor gases (Ström et al., 2003). Analysis of air mass
patterns for this region show that the shift in the size distributions is also accompanied by a change
of source areas, with a dominance of Eurasian source areas in winter and North Atlantic air during
summer (Tunved et al., 2013).
Ultrafine particle measurements including aerosol size distributions were also conducted from ice
breaker cruises such as from the Swedish ice breaker Oden (Bigg and Leck, 2001; Covert et al.,
1996; Heintzenberg and Leck, 2012; Leck and Bigg, 2005; Tjernström et al., 2014) and the
Canadian Ice breaker Amundsen (e.g. Chang et al., 2011). Chang et al., (2011) use model
calculations to show that the appearance of ultrafine particles can be explained by nucleation and
growth attributed to the presence of high atmospheric and oceanic DMS concentrations measured
at the same time. The Oden expeditions focus on the pack-ice-covered high Arctic, mainly north
of 80N and also confirm the frequent presence of an UFP mode (e.g. Covert et al., 1996). The
observations from the Oden cruises offer evidence that UFP in the inner Arctic might originate
from primary sources (e.g. Heintzenberg et al., 2015; Karl et al., 2013). This is motivated by three
main observations. First, a lack of sulfuric acid components in collected 15-50 nm particles (Leck
and Bigg, 1999). Second, Leck and Bigg (2010) highlight that nucleation events in the high Arctic
do not follow the classical banana shaped growth curve (Kulmala et al., 2001) but enhanced levels
of ultrafine particles rather appear simultaneously in distinct size ranges (Karl et al., 2012). Third,
the fact that such events cannot be modelled with empirical nucleation mechanisms for the
extremely low DMS concentrations in this region (Karl et al., 2013). As a primary source marine
microgels are suggested that might become airborne via the evaporation of fog and cloud droplets
(Heintzenberg et al., 2006; Karl et al., 2013).
So far most studies that include size distribution measurements in the summertime Arctic were
conducted from ground stations or ship cruises. To date there are only two studies that asses the





altitude dependence of the size distribution:, i.e. one in the area of Svalbard (Engvall et al., 2008)
and one from the Oden performing vertical profiles with a helicopter (Kupiszewski et al., 2013).
In this study we present data from aerosol size distribution measurements taken from an aircraft
during a three week period in July 2014 in the high Arctic area of Resolute Bay, Nunavut, Canada.
The flights focused on vertical profiles from as low as 60 m above the ground up to 3km, as well
as on low-level flights above different terrain such as fast ice, open ocean, polynyas and clouds.
We focus especially on UFP (5-20 nm) and address the following questions: What are the
concentrations of UFPs in the Arctic summertime, and what is their vertical distribution? What
are the environmental conditions that favour occurrence of UFPs? And, is there evidence for
growth of UFP to CCN sizes? Aside from the studies conducted near Svalbard, we believe this is
the first aircraft study in the high Arctic to systematically address these specific questions. This
work provides a comprehensive picture of UFPs observed during the campaign whereas a prior
publication from Willis et al., (2016) detailed one UFP formation and growth event observed over
Lancaster Sound.

**2   Experimental**
**2.1   Sampling Platform Polar 6**
The research aircraft Polar 6 owned by the Alfred Wegener Institute, Helmholtz Center for Polar
and Marine Research, Bremerhaven, Germany served as the sampling platform. The Polar 6 is a
converted DC-3 airplane (Basler BT-67) modified to work under extreme cold weather conditions.
An advantage of the plane is that flights at very relatively low speeds and altitudes (< 60 m a.g.l.)
are possible. The cabin of the aircraft is non-pressurized. We maintained a constant survey speed
of approximately 120 knots (222 km h$^{-1}$) for measurement flights at constant altitude, and ascent
and descent rates of 150 m min$^{-1}$ for vertical profiles. Instruments and measurements specific to
this paper are described below.




### 140  2.1.1 Inlets

Aerosol was sampled through a stainless steel inlet mounted to the top of the plane and ahead of
the engines to exclude contamination. The tip of the inlet consisted of a shrouded diffuser that
provided nearly isokinetic flow. Inside the cabin the intake tubing was connected to a stainless
steel tube (outer diameter of 2.5 cm, inner diameter of 2.3 cm) that carried the aerosol to the back
of the aircraft where it was allowed to freely exhaust into the cabin so that the system was not
over-pressured. The stainless steel tube functioned as a manifold, off which angled inserts were
used to connect sample lines to the various instruments described below. In-flight air was pushed
through the line with a flow rate of approximately 55 L min$^{-1}$ determined by the sum of the flows
drawn by the instruments (35 L min$^{-1}$), plus the flow measured at the exhaust of the sampling
manifold (20 L min$^{-1}$).  A flow of 55 L min$^{-1}$ was estimated to meet nearly isokinetic sampling
criteria at survey speed and the transmission of particles through the main inlet was approximately
unity for diameters between 20 nm to 1 µm.
Trace gases (CO and $H_2O$) were sampled through a separate inlet made of a 0.4 cm (outer diameter)
Teflon tube entering the aircraft at the main inlet and exiting through a rear-facing 0.95 cm exhaust
line that provided a lower line pressure. The sample flow of approximately 12 L min$^{-1}$ was
continuously monitored.

### 157  **2.2  Instrumentation**

### 158  2.2.1 Meteorological parameters and state parameters

Aircraft state parameters and meteorological measurements were performed with an AIMMS-20
manufactured by Aventech Research Inc. at a very high sampling frequency (>40Hz). The
AIMMS-20 consists of three modules: (1) The Air Data Probe that measures the three-dimensional
aircraft-relative flow vector (true air speed, angle-of-attack, and sideslip), and turbulence with a
three-dimensional accelerometer. As well, temperature and humidity sensors are contained within
this unit and provide an accuracy and resolution of 0.30 and 0.01 C for temperature and 2.0 and
0.1% for relative humidity measurements. (2) An Inertial Measurement Unit that consists of three
gyros and three accelerometers providing the aircraft angular rate and acceleration; (4) A Global
Positioning System for aircraft 3D position and inertial velocity. Horizontal and vertical wind





speeds were measured with accuracies of 0.50 and 0.75 m s$^{-1}$, respectively. The high frequency
raw data were processed to 1Hz resolution. Further details of the AIMMS including data
processing can be found in (Aliabadi et al., 2016a).

### 2.2.2   Aerosol physical and chemical properties

Particle number concentrations and particle size distributions were measured with a TSI 3787
water-based ultrafine Condensation Particle Counter (UCPC), a Droplet Measurement Technology
(DMT) Ultra High Sensitivity Aerosol Spectrometer (UHSAS) and a Brechtel Manufacturing
Incorporated (BMI) Scanning Mobility System (SMS) coupled with a TSI 3010 Condensation
Particle Counter (CPC). The UCPC detected particle concentrations of particles larger than 5nm
in diameter with a time resolution of 1 Hz. The flow rate was set to 0.6 L min$^{-1}$. The particle
concentrations measured by the UCPC are referred to as N$_{tot}$ hereafter.
The BMI SMS was set to measure particle size distributions from 20nm to 100nm with a sample
flow of 1 L min$^{-1}$ and a sheath flow of 6 L min$^{-1}$. The duration of one scan was 40 s with a 20 s
delay time before each scan resulting in a time resolution of 1min.  The UHSAS performed size
distribution measurements from 70 nm – 1 µm at a time resolution of 1 Hz with a sample floe rate
of 55 cm$^3$ min$^{-1}$. Details of the calibrations and instrument inter-comparisons performed prior and
during the campaign are described in detail in Leaitch et al. (2016).
Cloud condensation nuclei (CCN) were measured with a DMT CCN Counter (CCNC). The CCNC
was operated behind a constant pressure inlet that was set to 650 hPa. The nominal supersaturation
was held constant at 1%. Calibrations prior and during the campaign (for details see Leaitch et al.
2016) showed that a nominal supersaturation of 1% at the reduced pressure translated into 0.6%
effective supersaturation.
Cloud droplet sizes from 2-45 µm were measured using a wing mounted Particle Measuring
System (PMS) FSSP 100. In this study these data are only used to identify periods when the aircraft
was flying in cloud. To avoid possible artefacts produced from shattering of cloud droplets at the
aerosol inlet, data from in-cloud times are discarded for the purposes of this study.
A DMT Single Particle Soot Photometer (SP2) was deployed to measure refractory black carbon
(rBC) number and mass concentrations. We refer to rBC mass concentrations as an indication of



pollution influence. Calibrations with Aquadag soot were performed prior to and during the
campaign. The lower size limit of detection of rBC particles by the SP2 was approximately 80nm.
Sub-micron aerosol composition was measured with an Aerodyne high-resolution time-of-flight
aerosol mass spectrometer (HR-ToF-AMS; e.g. DeCarlo et al., 2006). A detailed description of
the instrument is found in Willis et al. 2016. The main purpose of the instrument was to measure
non-refractory particulate matter such as sulphate, nitrate, ammonium, methane sulfonic acid
(MSA) and the sum of organics. Lower detection limits were 0.009, 0.008, 0.004, 0.005 and 0.08
$\mu g\ m^{-3}$, respectively, and ~70 nm with respect to size in case of ammonium nitrate particles.

### 205 2.2.3 Trace gases

Carbon monoxide (CO) was measured with an Aerolaser ultra-fast carbon monoxide monitor
model AL 5002 based on VUV fluorimetry, employing the excitation of CO at 150 nm. In-situ
calibrations were performed during flight at regular intervals (15 – 30 min) using a NIST traceable
CO standard with zero water vapor concentration. CO mixing ratios were used as a relative
indicator of aerosol influenced by pollution sources.
Water vapour ($H_2O$) measurements were based on infrared absorption using a LI-7200 enclosed
$CO_2/H_2O$ Analyzer from LI-COR Biosciences GmbH. The measurement uncertainty is $\pm$ 15 $ppm_v$.
$H_2O$ mixing ratios were used to calculate relative humidity with pressure and temperature
measured by the AIMMS-20.

### 215 **2.3 Data analysis and nomenclature of particle size data**

All particle data were averaged to 1 min intervals to match the time resolution of the BMI SMS.
Particle concentrations within different size intervals were calculated. The notation $N_{a-b}$ is used;
"a" gives the lower limit and "b" the upper limit of the calculated size interval. The BMI SMS was
used to determine concentrations of particles from 20-90 nm diameter, and concentrations of
particles larger than 90 nm diameter were determined by the UHSAS. If the size interval is
expressed as $N_{>a}$ the upper limit is given by the detection limit of the UHSAS (1µm). Additionally,
particle concentrations from 5-20 nm (short: $N_{5-20}$) were obtained by subtracting particle





concentrations measured by the BMI SMS and by the UHSAS from the $N_{tot}$ as determined by the
CPC. The $N_{5-20}$ are also referred to as ultrafine particles (UFP) in this study.
In order to obtain vertical profiles the data were averaged within altitude intervals. Up to 500 m
where 50% of the flight time took place, the averaging interval was 50 m (50-100 m, 100-150 m
etc.). Above 500 m, where the data coverage was less, the data were averaged in 100 m altitude
bins. The altitude that is reported for a certain interval is given by the average altitude within this
intervaland not the middle altitude of the bin interval. Data points in the vertical profiles are
therefore not necessarily equidistant but better reflect the actual flying altitude.
Average size distributions were obtained by simply averaging each bin for the desired time and
altitude range. The size distributions measured by the BMI SMS were used for particle sizes from
20-90 nm, and the distributions at larger sizes are taken from the UHSAS. All particle
concentrations are expressed for ambient pressure conditions, i.e. they have not been adjusted to
standard temperature and pressure conditions. The $N_{5-20}$ referred to as UFP are added to the size
distributions as additional bin assuming a bin width of 15 nm (from 5-20 nm) with the mid diameter
of 12nm.

### 239    2.4  FLEXPART-WRF Simulations

We used FLEXPART-WRF (Brioude et al., 2013, website: flexpart.eu/wiki/FpLimitedareaWrf)
simulations run backwards in time to analyse the origins of air masses sampled along the flight
tracks. FLEXPART-WRF is a Lagrangian particle dispersion model based on FLEXPART (Stohl
et al., 2005). Meteorological information is obtained from the Weather Research and Forecasting
(WRF) Model (Skamarock et al., 2005). FLEXPART-WRF outputs retroplume information such
as the residence time of air (over a unit area) prior to sampling. Residence times were integrated
over the entire atmospheric column and 7 days backward in time. FLEXPART-WRF was run in
two ways.  First, one FLEXPART-WRF was completed for each flight using particle releases every
2 minutes along the flight track (100 m x 100 m x 100 m centered on the aircraft location) to
produce potential emissions sensitivities (PES) that represent the average airmass origin for each
flight. Second, separate runs were completed for points (every 10 minutes) along the flight track
(100 m x 100 m x 100 m, 60 second release duration) in order to study different airmasses
measured during the same flight. A more detailed description of the model as used for NETCARE
2014 is provided by Wentworth et al. (2015).

## 2.5   Study area and flight tracks

From July 4th to July 21st, 2014 eleven flights were conducted out of Resolute Bay (74.7 N, 95.0
W). In Figure 1 a compilation of all flight tracks on a satellite image is shown. The satellite picture
was taken on July 4th, 2014 and reflects the situation of the region during period I (July 4 to July
12). Resolute Bay proved to be an ideal location for this study as we had access to both open ocean
and ice covered regions. Additionally two polynyas were located north of Resolute Bay within the
reach of our aircraft. Flights ranged between 4-6 hours. The flights covered two main areas:
Lancaster Sound east of Resolute Bay and the area north of Resolute Bay where two polynyas
were located. The flights south of Resolute Bay in Lancaster Sound concentrated around the ice
edge.
The ice/water coverage visible on the satellite picture is representative for the area during the first
period. As can be seen, the ice edge was situated about 150 km east of Resolute Bay. It is clearly
visible in the satellite image as a sharp line. The transition from a completely ice covered region
to open ocean was very abrupt during the first period. Only after a period of bad weather with high
winds did the ice edge become less clear, and the region starting about 80 km east of Resolute Bay
to about 200 km east was covered by fractured ice.
Roughly 50% of the flight time was within the inversion layer, and 50% was in the free troposphere
conducting altitude profile flights. A considerable amount of time was spent at 2800 m as this was
the preferred altitude when travelling to a certain area. When clouds were present, the aircraft
sampled them by slant profiling through the cloud, in the case clouds were above the boundary
layer, or, in case clouds were within 200 m of the surface, by descending into the cloud as low as
possible. Aerosol observations while inside cloud are excluded from the analysis here due to
potential artifacts from droplets shattering on the outside inlet.





## 3   Meteorological and atmospheric conditions


Meteorological conditions changed over the course of the campaign. Similar conditions were
encountered during the first part of the campaign (July 4th – July 12th, 6 flights), referred to as the
"Arctic air mass period" because air masses from within the Arctic dominated and the atmosphere
showed structures typical for the Arctic such as a low boundary layer height with thermally stable
conditions, indicated by a near surface temperature inversion, and frequent formation of low level
clouds. At this time Resolute Bay was under the influence of high pressure systems. Clear sky with
few or scattered clouds and low wind speeds dominated. Conditions changed starting from July
13th when the region was influenced by troughs of a low pressure system located to the west above
Beaufort Sea, which eventually passed through Resolute Bay on July 15th bringing along humidity,
precipitation and fog. Intense fog and low visibility impeded flying from July 13th to July 16th. A
short good weather window in which the fog dissipated permitted flying again on July 17th
(referred to as "transition day"; one flight) just before Resolute Bay came under influence of a
pronounced low pressure system located to the south with its center around King William Island
(69.0 N, 97.6 W). The last campaign days (referred to as "southern air mass period", three flights)
were characterised by the influence of this pronounced low pressure system bringing air masses
from the south and providing higher wind speeds, an overcast sky and occasional precipitation.
Vertical profiles of median temperature, relative humidity (RH), wind speed, CO and $N_{tot}$ (Figure
2) illustrate median atmospheric conditions during the measurement flights. Prominent features
representing the trend of each period and reflecting the general meteorological situation will be
described here, details discussed in the respective sections. The Arctic air mass period was
characterized by frequent thermally stable conditions within the near surface layer , representing
typical conditions during the Arctic summertime (Aliabadi et al., 2016a; Tjernström et al.,
2012)The median temperature profiles show that on average the boundary layer reached up to
~300 m with a temperature increase of about 5 C. In this paper we will refer to this part of the
atmosphere as the boundary layer (BL) and to the air masses above as the free troposphere (FT).
A BL height of 300m corresponds well to the boundary layer height of 275 +/- 164 m estimated
by (Aliabadi et al., 2016a) using the method of bulk Richardson number (Aliabadi et al., 2016b)
and a critical bulk Richardson number of 0.5, using data from radiosondes launched at Resolute



Bay and the Amundsen icebreaker, which also performed research operations in Lancaster sound
during the campaign period.
Within the BL particle concentrations spanned over a wide range of concentrations (max $N_{tot}$:
~10000; median values: ~150 to ~1700 cm$^{-3}$). Highest $N_{tot}$ occurred during the Arctic air mass
period, while $N_{tot}$ was constantly low within the lower atmosphere on the transition day. Median
temperatures near the surface ranged from -1-5 C during the Arctic air mass period, largely
depending on the terrain below (e.g. ice or open water) and were clearly higher when the southern
air masses arrived (e.g. at the "surface": 4 C and 7 C, respectively). The higher temperatures
coincide with the influence of low pressure systems bringing warmer air masses from the west and
south and additional higher wind speeds providing a better mixing of the atmospheric layers (5.6
ms$^{-1}$ vs 12m$^{-1}$ near the surface). CO mixing ratios were extremely low during the Arctic air mass
period (median: 78.3 ppb$_v$) and on the transition day (median: 83.4 ppb$_v$) indicating pristine air
masses that had not recently been affected by pollution or biomass burning sources. The low CO
values are representative of background summer conditions (Law et al., 2014). During the southern
air mass influence CO mixing ratios clearly increased (median: 95.0 ppb$_v$) confirming a change in
air mass and suggesting possible influences by pollution sources and wild fires in the North West
Territories (Supplementary Figure 2). Relative humidity profiles show that the near surface layer
of the atmosphere was very moist with RH > 80 % during all periods.

**4    Results and Discussion**
**4.1    Ultrafine particle events**
4.1.1  Frequency of ultrafine particle events
Throughout the campaign we observed large variability in particle concentrations (Figure 3). We
observed not only very clean air masses with $N_{tot}$ of a few tens cm$^{-3}$ (with the lowest 1-second
value of 1 cm$^{-3}$), but also concentrations as high as a few thousands per cm$^{-3}$ (with the highest
value of 10 000 cm$^{-3}$). The highest and lowest concentrations were measured within the BL (Figure
3b). Above the BL (Figure 3b) particle concentrations were relatively constant where 60 % of the





time concentrations were between 200-300 cm$^{-3}$ (for a discussion of the average size distribution
see sections 4.1.2 - 4.1.4). Especially during the Arctic air mass period (Figure 2) the atmosphere
was characterized by a strong contrast between the BL and the FT.
UFP were very frequently present within the BL in high concentrations (Figure 3c). Bursts of $N_{5-20}$
$> 2000/cm^3$ were observed over polynas, in Lancaster Sound and south of Resolute Bay. The $N_{5-}$
$_{20}$ was higher than 200 cm$^{-3}$ during 65 % of the time. Indeed, high $N_{tot}$ was mainly driven by UFP
(as can be seen by comparison of black dots indicating high $N_{tot}$ in Figure 3c and high UFP in
Figure 3d). Whenever $N_{tot}$ is greater than 2000 cm$^{-3}$, UFP was at least larger than 1000 cm$^{-3}$. This
is also illustrated by the ratio of UFP/$N_{tot}$ (Figure 3e). A ratio of 0 means that no UFP were present,
while a ratio of 1 means that only UFP were present. Within the boundary layer 32 % of the time
the size distribution was dominated by UFP (ratio > 0.5).
The frequent presence of UFP agrees well with other studies made during the Arctic summertime
at several locations, such as at the ground stations in Ny Alesund and Zeppelin (Ström et al., 2009;
Tunved et al., 2013), at Alert (Leaitch et al., 2013), also from ship-based observations (Chang et
al., 2011; Covert et al., 1996; Heintzenberg et al., 2006). However, such a frequent presence of an
UFP mode (65 % of the time > 200 cm$^{-3}$) in the BL is unique to this study. Possible reasons for the
higher occurrence of UFP might be the combination of the proximity of open ocean (providing a
source of UFP or precursor gases), favourable meteorological conditions (sunny weather, inversion
layer with cloud formation) and very clean air masses with low condensation sinks. Calm weather
conditions may have been another factor. The highest concentrations of UFP were measured at
lower wind speeds (< 5 m s$^{-1}$; Supplementary Figure 1), while lower UFP concentrations (1000
cm$^{-3}$) were found at high wind speeds (>12 ms$^{-1}$) suggesting a dilution effect of the wind.
In the following sections, the vertical distribution of UFP and the size distributions are discussed
in relation to meteorological conditions during the three distinct periods that characterized this
campaign.



### 4.1.2 Arctic air mass period: July 4th to July 12th

During this first period the study area was under the influence of a high pressure system. As illustrated by FLEXPART-WRF results (Figure 4a and 4b), air masses were either coming from the North extending to the East in the Arctic Ocean or from the East passing over the open ocean in Lancaster Sound and Baffin Bay. Both examples indicate that air masses resided within the Arctic region at least 5 days prior to sampling. This is true for all flights during this period. The very low CO mixing ratios (78 ppb$_v$, see Figure 2) and average BC mass concentrations of 3 ng m$^{-3}$ (not shown) confirm that air masses were very clean and without recent influence from pollution sources. As discussed in section 3, temperature profiles indicate thermally stable conditions in the lowest layers with near-surface temperature inversions. During almost all vertical profiles we observed temperature inversions of about 4-6 C near the surface. Such an atmospheric structure i.e. a shallow boundary layer is typical for the Arctic summertime (e.g. Aliabadi et al., 2016a; Tjernström et al., 2012).

The Arctic air mass period was characterized by a very sharp contrast between the BL and the FT in terms of particle number concentrations and sizes (Figure 5). The BL was characterized by a prominent layer of UFP from the surface to 300 m (Figure 5a). The height of the UFP layer coincides with the average height of the temperature inversion for this period (see temperature profile Figure 2) and indicates that air masses were stably layered enabling only little exchange with the FT. This is supported by the observed lower turbulent mixing (i.e. turbulent kinetic energy) from boundary layer to the free troposphere during the campaign (Aliabadi et al., 2016a).

During this period we measured the highest concentrations of UFPs with the one minute average up to 5300 cm$^{-3}$. On a typical flight several bursts (concentrations suddenly rising from close to zero to several hundreds and thousands cm$^{-3}$) of high UFP concentrations were encountered in the SIL. These "bursts" lasted from a few seconds to several minutes, corresponding to a spatial extent of several hundreds of meters to dozens of kilometers. The large spatial variability is also illustrated by the frequency distribution of UFP in the BL shown in Figure 5c. E.g. 40 % of the time concentrations of UFP were larger than 200 cm$^{-3}$, 11 % of the time larger than 1000 cm$^{-3}$ and 3% of the time even larger than 2000 cm$^{-3}$. Interestingly the highest concentrations are not found right above the surface (i.e. at the lowest flight altitude of around 60 m) but at a higher altitude



(140-170 m). In part, that may be due to the fact that very high concentrations of UFP were
measured above the top of low-level clouds, to be discussed in Section 4.3. The average $N_{20-40}$
follow a similar trend as the UFP and show also an increase in concentrations within the BL. In
contrast, concentrations of larger particles ($N_{>40}$, $N_{>80}$, $N_{>150}$) are minimal at the lowest altitude
resulting in a very clean BL with low surface areas (~5 $\mu m^2$ $m^{-3}$ and lower). Within the FT particle
concentrations were surprisingly uniform and concentrations of UFP were less than 50 $cm^{-3}$ up to
1200 m and ~10 $cm^{-3}$ above.
In figure 5b the average and median size distribution is shown and illustrates that increases in only
UPF above background levels, without larger particles sizes, were very frequent. However, at
times high concentrations of particles extended up to about 40nm. This might indicate that at those
times particles experienced growth to larger sizes. Such a case will be discussed in Section 4.3.
It should also be noted that occasionally a mode of particles larger than 400 nm was present,
associated with open water (see Section 4.2) and therefore most likely the product of primary
oceanic emissions.

### 4.1.3  Transition day on July 17th

July 17th marks the transition from a dominance of Arctic air masses to a clear influence from
southern air masses. The transition day consist of only one flight in the area of Lancaster Sound,
which, however, is especially interesting as we observed a very "clean" air mass (low
concentrations of large particles, e.g. $N_{>40}$: 60-100 $cm^{-3}$, see Figure 6), probably as a result of cloud
processing and scavenging during the days before that occurred when flying was impossible
because of intense fog and cloud formation at Resolute Bay.
On this day the low pressure system situated to the west was bringing air masses from the west
along the Canadian and Alaskan coastline (Figure 4c). The temperature profile shows an inversion
between 650-1000 m possibly indicating a change in air mass. CO mixing ratios (83 $ppb_v$) and BC
mass concentrations (3 ng $cm^{-3}$) are still quite low indicating mostly Arctic background conditions.
Again we observed a mode of UFP within the near surface layer (Figure 6b) determined by
occasional bursts of UFP with concentrations up to 1400-1900 $cm^{-3}$. UFP of 200 $cm^{-3}$ or more
were observed about 20% of the time (Figure 6c) with an average concentration of 240 $cm^{-3}$.





Concentrations of larger particles ($N_{>40}$, $N_{>80}$, $N_{>150}$) increased sharply at about 700 m coinciding
with the temperature inversion. The very low concentrations of larger particles ($N_{>40}$: ~70 cm$^{-3}$)
below the temperature inversion are very similar to the conditions encountered within the BL
during the previous period. Prior to the transition day the air mass below 700 m was likely cleaned
by the clouds and fog that had covered the area during the days before the flight. Only a few hours
before the start of the flight, the fog had dissipated and also the terminal aerodrome forecast
reported a lift of the cloud base from < 100 m to around 450 m. Median and average size
distributions indicate a minimum at around 65 nm that might be the result of cloud processing
(Hoppel et al., 1994), consistent with the Arctic observations of Heintzenberg et al., (2006).
### 4.1.4  Southern air mass period: July 19$^{th}$ – July 21$^{st}$
During this period the region was under strong influence of a low pressure system centered south
of Resolute Bay. FLEXPART-WRF air mass trajectories (Figures 4d and 4e) indicate a prevalence
of air masses from the south potentially affected by wild fires (see Supplementary Figure 2). At
the beginning of this period on July 19$^{th}$ (Figure 4d), air mass trajectories suggest the strongest
influence from the south while towards the end of the period on July 21$^{st}$ (Figure 4e), FLEXPART-
WRF indicates that southern air masses mixed with air masses coming off Greenland. Near surface
temperatures were higher than during the previous periods (Figure 2) and temperature inversions
were less pronounced (2-4 C) and not observed at all locations suggesting a less stable lower
atmosphere. On July 19$^{th}$ we encountered the highest wind speeds in the lower atmosphere (16 m
s$^{-1}$ within the near surface layer and 20 m s$^{-1}$ slightly above) and also RH was highest (near the
surface 91%) and did not drop below 80% throughout the vertical atmosphere. CO mixing ratios
were higher than during the prior periods suggesting that the air was at times influenced by
pollution or biomass burning.
UFP were observed less frequently than during the Arctic air mass period and in lower
concentrations (Figure 7). Bursts of UFP above 1000 cm$^{-3}$ occurred only at three locations, all
during the flight on July 21$^{st}$. Average UFP concentrations were only approximately 190 cm$^{-3}$.
UFP concentrations of 200 cm$^{-3}$ or higher were detected 31 % of the time below 300m (Figure 7c).





The southern air mass period clearly shows different aerosol characteristics within the near surface
layer than compared to the Arctic air mass period and the transition day. Average concentrations
of particles larger than 40 nm were highest within the boundary layer and decreased with altitude
(Figure 7a). This is in sharp contrast to the very clean (with respect to particles larger than 40 nm)
boundary layers observed before. Whereas concentrations of particles larger 40nm were around
~100 cm$^{-3}$ and lower during both prior periods, they were as high as 300 cm$^{-3}$ for this period. Even
large accumulation mode particles ($N_{>150}$) averaged ~50 cm$^{-3}$ (compared to 10 cm$^{-3}$ for both
previous periods). The average size distribution (Figure 7b) illustrates this in detail: both median
and average size distributions show a pronounced mode of particles larger than 500 nm within the
BL. Primary emissions from the sea spray promoted by the higher surface wind speeds (see Figure
2) are likely a factor contributing to the larger particles.
During the southern air mass period, three important factors had changed compared to both prior
periods. (1) Air mass back trajectories had clearly shifted to the south and potentially transported
emissions from wild fires located in the Northwest Territories (Supplementary Figure 2) into the
region, which might mix into the boundary layer. (2) The Amundsen ice breaker was present in
Lancaster Sound and acted as a local pollution source (Aliabadi et al., 2016c). (3) Wind speeds
were higher and the ocean was visibly turbulent with breaking waves that might enhance primary
oceanic aerosol emissions. The increased condensation sinks from these potential sources in
combination with other factors (e.g. reduced sun light) and relatively low residence times of air
masses within the boundary layer (compared to the Arctic air mass period) may explain the
relatively low and infrequent concentrations of UFPs.
Within the FT the size distributions shows a bimodal character with a minima at around 65 nm,
which possibly indicates that the air masses had experienced cloud processing. Hoppel et al.,
(1994). This is very likely, given the presence of the low pressure system bringing moist (see RH
~80% in Figure 2) and warmer air masses. Towards smaller particles the size distribution flattens
out (median size distribution) with occasional increased concentrations of particles below ~40 nm
(average size distribution). This bimodal size distribution is clearly different from the size
distribution during the Arctic air mass period when dry air masses from within the Arctic
dominated.



## 4.2  UFP occurrence above ice versus water


We investigated the potential influence of different terrain on the occurrence of UFP by examining
in detail the time periods when we were flying at altitudes at or below 300 m during the Arctic air
mass period. We distinguish between three underlying surfaces: ice covered areas (including ice
edge and ice covered with melt ponds), open ocean (including polynyas), and low-level clouds
(including both cloud above water and cloud above ice). Here we point out that the case "cloud"
does not include in-cloud flight times but only flight periods when above cloud top without actually
entering the cloud (confirmed by a zero signal in a liquid cloud probe, FSSP100). An altitude of
500 m was chosen to include time periods when we were flying above low-level clouds and to
capture mostly flights within the boundary layer where a local influence of the terrain below was
likely. During the Arctic air mass period the terrain was characterized by a clear separation
between ice and open water. East of the ice edge Lancaster Sound was completely ice free, while
west of the ice edge the ocean was seamlessly covered by ice (see satellite picture in Figure 1).
All profile above different terrain show unique features (Figure 8). Above ice the highest
concentrations of UFP (average: 405 cm$^{-3}$) were found nearer the surface (70 m). In the BL over
open water as well as just above cloud, the average number concentrations were higher (1025 cm$^{-3}$
and 1533 cm$^{-3}$, respectively) and were found at slightly higher altitudes (140 m and 170 m,
respectively). The cloud case is particularly interesting as $N_{20-40}$ particles show an increase at the
same time as the UFP suggesting that above cloud UFP do not only form but also grow to larger
sizes. In the over-ice and over-open water cases the $N_{20-40}$ show a pattern more similar to the larger
particles indicating that their appearance is governed by the same processes as the larger ones. For
example, the open water case shows an increase for all particle sizes larger than 20 nm right at the
surface that might indicate growth of UFP near the surface. In contrast, UFP reach their highest
concentration when all other particle sizes reach a minimum. An increased abundance of UFP at
lower surface areas supports the hypothesis that UFP form via nucleation of precursor gases.





### 4.3  Case study: July 8

The flight on July 8 provides a case study focussing on UFP above cloud and within the BL during the Arctic air mass period. We consider the altitude dependence of the UFP within the BL in relation to air mass history and the possible connection with cloud.

On this flight we first flew out into Lancaster Sound west of Resolute Bay, turned around and descended into the BL above ice. Here, we focus on the time period from 15:50 UTC (descent into the BL) to 17:20 UTC where we travelled from west to east and remained within the BL but stayed out of cloud ( Figure 9; see also Supplementary Figure 2 for the flight track shown on a satellite image). The later part of the flight focused on in-situ cloud properties and is discussed elsewhere (Leaitch et al., 2016). The weather was sunny with low level clouds starting around 150 km over ice and west of the ice edge in Lancaster Sound. The clouds had formed over the water and were blown over the ice where they partly dissipated (Leaitch et al., 2016). In the entire area the atmosphere was characterized by a surface temperature inversion extending vertically up to about 300 m with ~1 C near the surface and ~5 C at 300 m and was accompanied by decreasing relative humidity (Figure 9f). Local low-level winds were dominantly coming from the south to east sector (Figure 9g) and wind speeds were below 5 ms$^{-1}$.

UFP were present throughout the BL with the highest concentrations at the lower altitudes and decreasing concentrations towards the top of the BL (Figure 9b). In contrast, larger particles (e.g. $N_{>40}$) exhibit the opposite pattern, with lower concentrations at lower altitudes and higher concentrations at higher altitudes. Six locations from west to east (points A-F in Figure 9a) are used to illustrate the changing aerosol characteristics. Location A is situated well above the BL and at this point no UFP were present (detailed size distributions are shown in Supplementary Figure 4). At location B, the point at which we first entered the BL, an UFP mode (~370 cm$^{-3}$) was present at 60m, while UFP concentrations were lower at slightly higher altitudes (~80 cm$^{-3}$ at 230 m) such as location C. At the lower altitudes the UFP mode gradually increased as we approached the ice edge. The most striking observation is the steep increase in particle concentrations at about 60 km west of the ice edge (location D) at altitudes of 100-150 m. UFP increased to above 4000 cm$^{-3}$ at 150 m. At the same time $N_{20-40}$ concentrations showed a similar increase which was not the case before. At this point we were just above the thickened cloud layer. Notably, increased UFP



concentrations were limited to the vicinity of cloud top and decreased rapidly with increasing
altitude. Further east (after 1700 UTC, point F) we were restricted to above cloud top and close to
the top of the SIL and no peaks of aerosol concentrations were observed with the exception of just
before the ice edge (location E).
In order to interpret these observations, air mass histories at these locations were investigated with
FLEXPART-WRF (Figure 10) and indicate the following:
(1) To the west of Resolute Bay (point B) Lancaster Sound air masses had been mixed with air
masses from the North. This is also confirmed by the local wind directions indicating winds
coming from the Northwest sector (Figure 10a), and it is consistent with the associated change in
cloud. (2) Towards the top of the BL air masses had descended only very recently (< 3 h) into the
SIL from above the BL (Figure 10c point C and point F). (3) In contrast, deeper within the BL
such as at point ii and at point iv air masses had descended into the BL earlier (~20 h) before
arriving at the point of observation. In the case of point D, where we observed the largest mode of
UFP extending above 40nm, air masses had been travelling from the east exclusively over the open
waters in Lancaster Sound during the last day before arriving at the point of observation (Figure
10b, point D).
Aerosol composition shows a clear difference between the aerosol in the FT and the BL. The
aerosol sulphate rapidly decreases as we enter the BL around 16:00, while aerosol organic loadings
show an initial relative increase followed by an absolute increase towards the east (Figure 9c).
Within the BL aerosol organics and sulphate mass loadings show a pattern similar to $N_{>40}$ and
$N_{>80}$. Both decrease each time we descended deeper into the BL. However, at the same time the
organics-to-sulphate ratio indicates that the relative contribution of organics to aerosol mass
increases at lower altitudes and especially above cloud (Figure 9e). Well within the inversion layer
and in the vicinity of cloud top the aerosol was dominated by organics. At the same time also ratio
of MSA to sulphate was higher (Figure 9e), suggesting a marine biogenic influence of the aerosol
sulphur. The marine biogenic influence at the lower altitudes agrees well with the FLEXPART-
WRF simulations showing that air masses at this altitude had spent almost an entire day exposed
to the open waters in Lancaster Sound. For a more detailed discussion of organic aerosol and
growth see (Willis et al., 2016) and references therein.



Consistent with the higher organic content measured with the AMS, the single particle aerosol
mass spectrometer ALABAMA (Brands et al., 2011; Willis et al., 2016) detected a higher fraction
of trimethylamine (TMA)-containing particles for particles larger than 150 nm in diameter.
Gaseous TMA emissions from marine biogenic origin (Ge et al., 2011; Gibb et al., 1999) may have
additionally favored the subsequent growth of the freshly nucleated particles by condensation.
Another possibility may be uptake of TMA in the cloud phase (Rehbein et al., 2011) if the particles
have grown to sufficiently large sizes to be activated as CCN. Interestingly, compared to other
days these TMA-containing particles are smaller and to a lesser degree internally mixed with
potassium and levoglucosan which supports the hypothesis of ultrafine particles originating from
nucleation in a biogenic marine environment and subsequent growth. A detailed discussion of
TMA-containing particles observed during this campaign will be presented in (Köllner et al., 2016
in prep).
To explain these observations, we hypothesize that the smaller particle mode is formed by
nucleation and growth occurring within the BL and especially in cloud vicinity. UFP
concentrations near cloud top have been reported before (e.g. Garrett et al., 2002; Hegg et al.,
1990) and it is suggested that nucleation in near cloud regions is favoured by the low surface areas
in cloud scavenged air masses, moist air and a high actinic flux. Indeed, in cloud vicinity where
we observed an increase of UFP extending up to almost 50nm the conditions for nucleation and
growth are ideal: we speculate that the availability of precursor gases is provided by the long
residence time (~20h) of the air masses over open water. The very high organic loadings and MSA
to sulphate ratio likely indicate that the formation of these particles is driven by organic precursors.
Precursor gases such as DMS and volatile organic compounds (VOCs) are likely emitted by the
open ocean in Lancaster Sound (e.g. Ghahremaninezhad et al., 2016 in prep; Mungall et al., 2016).
At first sight the increase of UFP at point E seems to contradict these observations. The event
occurs at a point where the aircraft was clearly above cloud and close to the top of the BL, at a
location where no increases in UFP were observed before or after. However, at this point we were
in vicinity of Prince Leopold Island which is a bird sanctuary and many bird colonies nest at the
260m high cliff. FLEXPART-WRF shows that air masses to a large extent were directly coming
off the island (Figure 10, point E) suggesting a connection between the appearance of UFP and
possible emissions from the fauna of the island. The increase of particle phase ammonium, $NH_4$,
(Figure 9d) at the same time strongly supports this connection and nucleation of particles from
biogenic precursors emitted by bird colonies are documented (Weber et al., 1998; Wentworth et
al., 2015, Croft et al. 2016b).
Alternatively, it should be considered that evaporating fog and cloud droplets may also act as a
primary source of UFP (e.g. Heintzenberg et al., 2006; Karl et al., 2013; Leck and Bigg, 1999).
Karl et al., (2013) suggested a combined pathway that involves the emission of UFP by fog and
cloud droplets, together with secondary processes enabling growth of these particles. For our
observations we have no reason to assume that nucleation does not occur since conditions are ideal
but we cannot rule out that nanoparticles are emitted by the possibly evaporating cloud droplets
onto which gases then condense.
In conclusion the aerosol mass within the near surface layer is dominated by organics relative to
sulphate, while at just slightly higher altitude sulphate is clearly increased and increases further
above the inversion layer. A high organic content coincides with increases in UFP particles,
especially at times when also growth into the size range up to 50nm is indicated. Similarly the
MSA-to-sulphate ratio shows a peak at the lowest altitudes with maximum values in the vicinity
of clouds that coincide with a long residence time (~20h) of the air masses within the BL and
above open water. The data thereby suggest a marine biogenic influence of the aerosol within the
lower layers of the atmosphere. We note that similarly high levels of aerosol organics and MSA
were observed during the flight on July 12 associated with a NPF event and growth but in cloud-
free conditions Willis et al. (2016).
**4.4  CCN activity**
CCN concentrations were measured at a supersaturation of 0.6 %. The vertical profiles of CCN
concentrations (Figure 11a) show patterns similar to those of larger particles. In the very clean
boundary layer of the Arctic air mass period and the transition day CCN concentrations are equally
low (~70 $cm^{-3}$ and ~50 $cm^{-3}$, respectively). In contrast, during the southern air mass period average
CCN concentrations are amongst the highest observed during this campaign (>300 $cm^{-3}$). Within
the free troposphere CCN concentrations are surprisingly constant during the Arctic air mass



period ($120 \pm 27$ cm$^{-3}$) and more variable on the transition day ($92 \pm 46$ cm$^{-3}$) and the southern air
mass period ($103 \pm 67$ cm$^{-3}$). The constant CCN concentrations during the Arctic air mass period
correspond to the very uniform atmosphere dominated by aged aerosols we observed during this
period and to the more layered atmosphere influenced by southern air masses possibly
contaminated by biomass burning plumes during the later period. Correlations with $N_{>80}$ (Figure
11b) confirm that larger particles are a good approximation for these CCN concentrations. On
average CCN concentrations agree roughly to within $\pm$ 20 % of $N_{>80}$ suggesting that a diameter of
80 nm is a good approximation for the activation diameter of the aerosol at 0.6 % supersaturation.
However, it should be noted that slight differences between the 3 periods are indicated in the
correlation curves: during the Arctic air mass period the average activation diameters are smaller
than 80 nm, and during the southern air mass period they are larger than 80 nm. Assuming uniform
chemical composition throughout the particle size range, an activation diameter of 80 nm at 0.6 %
supersaturation indicates an aerosol much less hygroscopic than, for example, ammonium
sulphate; pure ammonium sulphate particles would activate at 40 m at 0.6 % supersaturation. For
the one specific event during which growth occurred (Willis et al., 2016), it was demonstrated that
high CCN concentrations coincide with elevated organic mass loading. The reduced
hygroscopicity of organic material (Petters and Kreidenweis, 2007) can explain the larger effective
activation diameter.

## 633    5   Discussion and Conclusions

This study presents airborne observations of ultrafine particles (UFP) during the Arctic
summertime. The study of Leaitch et al., (2016) for this same campaign has illustrated the
importance of small particles (20 – 100 nm) to cloud formation and thus climate in this region.
Eleven flights were conducted in July 2014 in the area of Resolute Bay situated in the middle of
the Canadian Archipelago. The location allowed access to open water, ice covered regions and
polynyas. Flights focused around the ice edge in Lancaster Sound (7 flights) including the open
waters to the east and the ice covered region to the west, and north of Resolute Bay around the
polynyas (3 flights). UFP were observed within all regions and above all terrains with the highest
concentrations encountered in Lancaster Sound above cloud and open water. UFP observations





were discussed in relation to different meteorological conditions (Arctic air mass period, transition

day, southern air mass period). It is shown that UFP occur most frequently (>65 % of the time)

and with the highest concentrations (up to 5300 $cm^{-3}$) during the Arctic air mass period when the

air is relatively clean and the BL thermally stable.

The frequent presence of UFP in the BL during the Arctic air mass period over open water and

low cloud suggests a surface source for the observed particles, such as the ocean. This is especially

true during the Arctic air mass period when the sampling region was experimentally found to be

pristine and not influenced by pollution, as confirmed by the FLEXPART-WRF simulations that

show air masses had resided within the Arctic region at least 5-7 days prior to sampling. During

this time UFP were restricted to the boundary layer and no UFP events were observed aloft, thereby

excluding that these UFP might form in the free troposphere and subside into the near surface layer

e.g. (Clarke et al., 1998; Quinn and Bates, 2011). At the same time we observed an extremely clean

BL (surface area of $N_{>40}$ ~5$\mu m^2 m^{-3}$). Low surface areas increase the probability of particle

formation via nucleation by reducing the surfaces for precursor gases, such as DMS or VOCs, to

condense on. Chlorophyll-a concentrations (Supplementary Figure 5) suggest there was a

relatively high level of biological activity of the ocean (such as phytoplankton blooms known to

produce DMS) throughout Lancaster Sound, to the east in Baffin Bay and in the open waters of

the polynyas during the time period of the study.

Measurements of gas phase DMS in Lancaster Sound performed from the Amundsen ice breaker

just a few days after the aircraft campaign described in this study show that DMS was ubiquitous

in the Lancaster Sound region (Mungall et al., 2016). Mixing ratios ranged from 4-1155 $ppt_v$ with

the highest value measured at the east end of Lancaster Sound and a median value of 186 $ppt_v$ (for

the entire study time and area including Baffin Bay and Nares Strait). DMS was also measured

from the Polar 6 aircraft with an offline technique and also shows a large variability within the BL.

Maximum mixing ratios of 110 $ppt_v$ were detected in the surface layer (Ghahremaninezhad et al.,

2016 in prep.), again confirming a marine influence in the boundary layer. The measured DMS

concentrations are above the nucleation threshold obtained by modelling performed in the study

of Chang et al. (2011) who concluded that DMS mixing ratios of $\geq$ 100 $ppt_v$ are sufficient to

account for the formation of hundreds of UFP when background particle concentrations are < 100



cm$^{-3}$. These conditions agree very well with our observations during Arctic air mass period, when
background particle concentrations ($N_{>40}$) within the boundary layer were low.
Relating observations of UFP to the surface below (ice, water, low-level cloud) during the Arctic
air mass period revealed that the highest UFP concentrations occurred above low-level cloud and
open water with averages of 1533 cm$^{-3}$ and 1025 cm$^{-3}$, respectively. Above low-level cloud in
addition to UFP also $N_{20-40}$ showed increased concentrations. This simultaneous increase in
concentrations suggests that UFP grow into the 40 nm size range.
Overall, the summertime Arctic is an active region in terms of new particle formation, occasionally
accompanied by growth. The value of the altitude profiles across a wide spatial extent, performed
for the first time in this campaign, is that they demonstrate that this activity is largely confined to
the boundary layer, and that the dominant source of small particles to the boundary layer does not
arise by mixing from aloft but most likely from marine sources. For future studies, the relative
impact of such natural sources of UFP needs to be evaluated with respect to potential new sources,
such as may arise with increasing shipping.

## Acknowledgements

The authors would like to thank a large number of people for their contributions to this work. We
thank Kenn Borek Air, in particular the pilots Kevin Elke and John Bayes and the aircraft engineer
Kevin Riehl. We are grateful to John Ford, David Heath and the University of Toronto machine
shop for safely mounting our instruments on racks for aircraft deployment. We thank Jim Hodgson
and Lake Central Air Services in Muskoka, Jim Watson (Scale Modelbuilders, Inc.), Julia Binder
and Martin Gerhmann (Alfred Wegener Institute, Helmholtz Center for Polar Marine Research,
AWI), Mike Harwood and Andrew Elford (Environment and Climate Change Canada, ECCC), for
their support of the integration of the instrumentation and aircraft. We gratefully acknowledge
Carrie Taylor (ECCC), Bob Christensen (U of T), Lukas Kandora, Manuel Sellmann and Jens
Herrmann (AWI), Desiree Toom, Sangeeta Sharma, Dan Veber, Andrew Platt, Anne Marie
Macdonald, Ralf Staebler and Maurice Watt (ECCC) for their support of the study. We thank the
Biogeochemistry department of MPIC for providing the CO instrument and Dieter Scharffe for his
support during the preparation phase of the campaign. The authors J.L. Thomas and K.S. Law



acknowledge funding support from the European Union under Grant Agreement n_ 5265863 –
ACCESS (Arctic Climate Change, Economy and Society) project (2012-2015) and TOTAL SA.
Computer simulations were performed on the IPSL mesoscale computer center (Mésocentre
IPSL), which includes support for calculations and data storage facilities. We thank the Nunavut
Research Institute and the Nunavut Impact Review Board for licensing the study. Logistical
support in Resolute Bay was provided by the Polar Continental Shelf Project (PCSP) of Natural
Resources Canada under PCSP Field Project #218614, and we are particularly grateful to Tim
McCagherty and Jodi MacGregor of the PCSP. Funding for this work was provided by the Natural
Sciences and Engineering Research Council of Canada through the NETCARE project of the
Climate Change and Atmospheric Research Program, the Alfred Wegener Institute, Helmholtz
Center for Polar and Marine Research and Environment and Climate Change Canada.



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




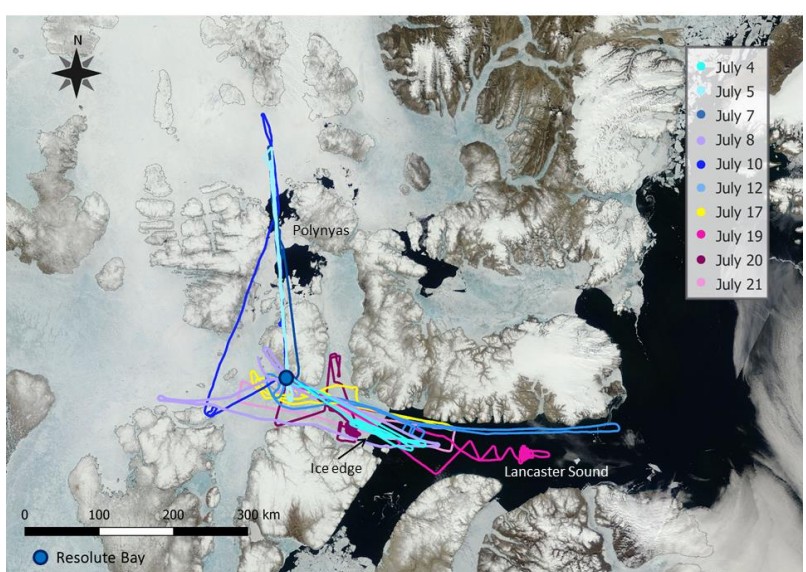


Figure 1. Compilation of all flight tracks plotted on a satellite image from July 4, 2014. The image
is taken from: https://earthdata.nasa.gov/labs/worldview.





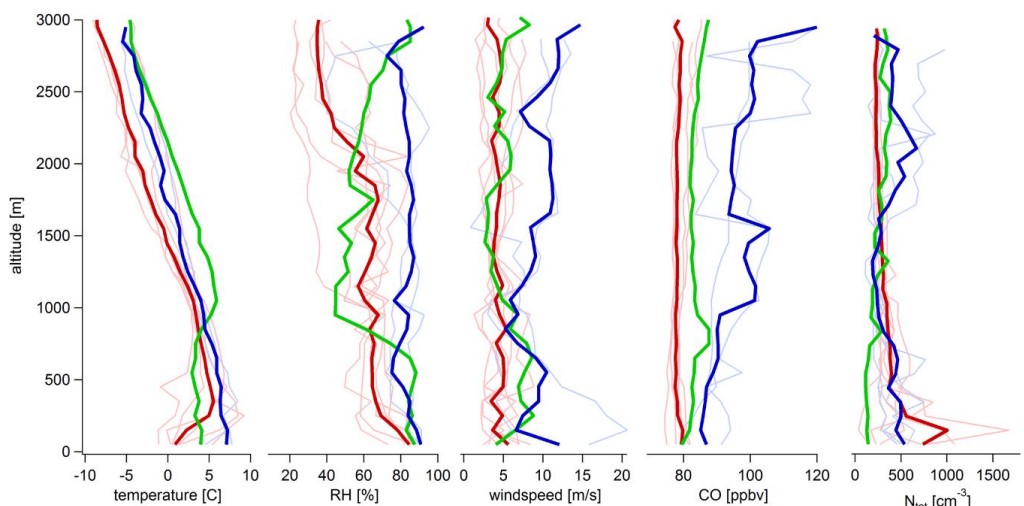


Figure 2. Median temperature, relative humidity (RH), wind speed, CO mixing ratio and $N_{tot}$
profiles for the Arctic air mass period (dark red), the transition day (dark green), and the southern
air mass period (dark blue). Median profiles for each flight are plotted in the background in the
corresponding light colours.



949
Figure 3. Flight tracks colour coded by particle concentrations. a.) Flight tracks within the boundary layer (50-300 m) colour coded by $N_{tot}$. b) Flight tracks within the free troposphere (300-3000 m) colour coded by $N_{tot}$. c) Flight tracks within the boundary layer (50-300m) colour coded by UFP. d) Flight tracks within the free troposphere (300-3000 m) colour coded by $N_{5-20}$. e) Flight tracks within the boundary layer (50-300 m) colour coded by the ratio of $N_{5-20}/N_{tot}$

955



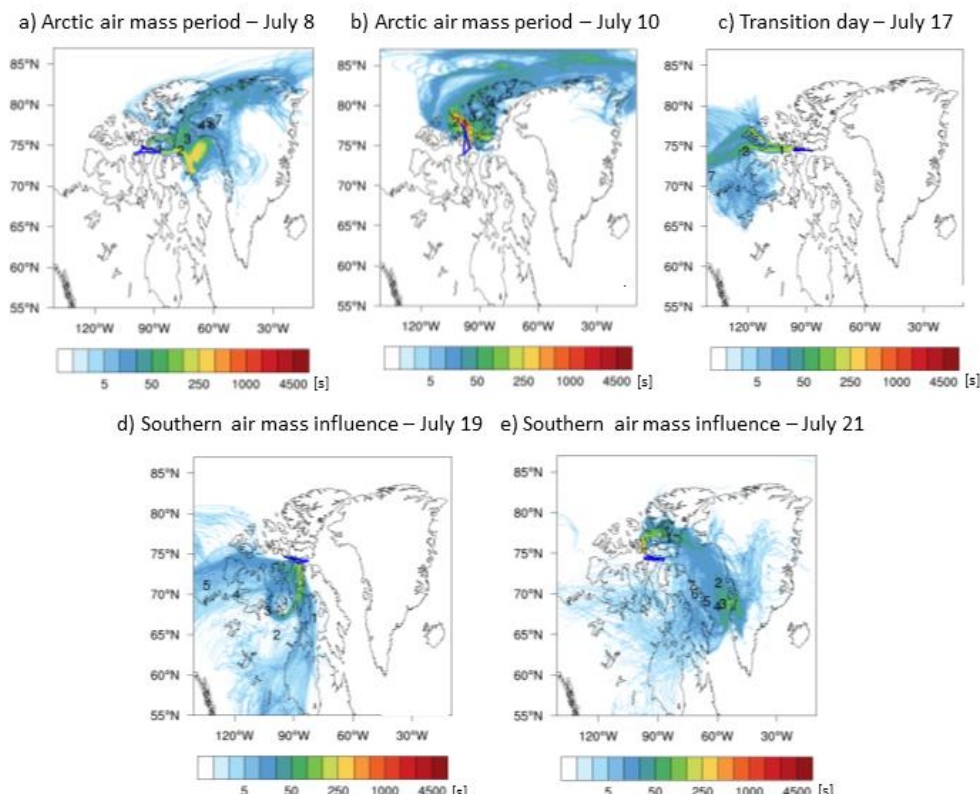

956

Figure 4. FLEXPART-WRF potential emissions sensitivities for each flight (using particle releases every 2 minutes along the flight track) that illustrate transport regimes during different periods of the campaign. The colour code indicates the residence time of air in seconds and the numbers represent the position of the plume centroid location in days prior to release (days 1-7).






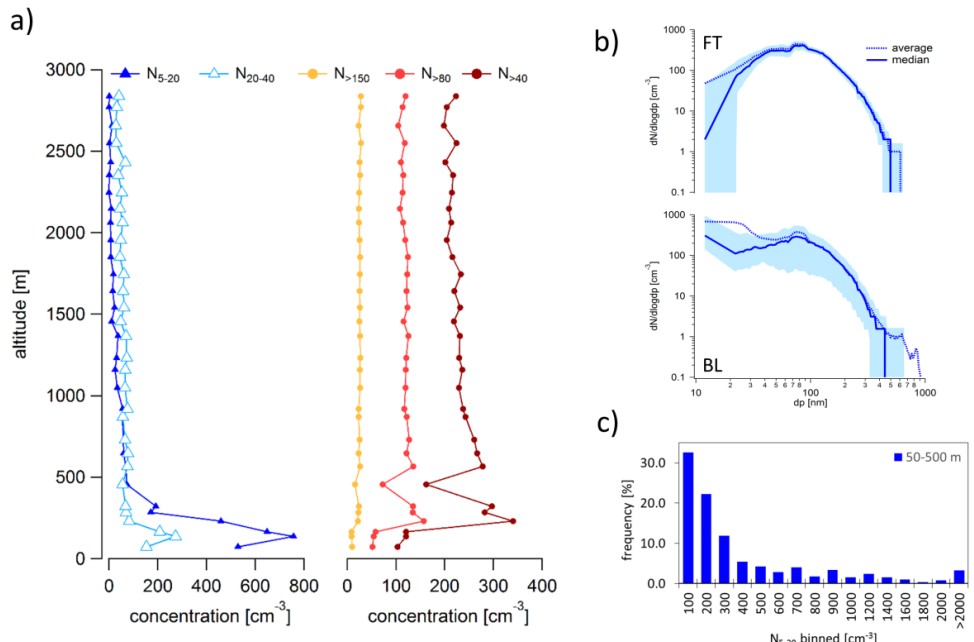


Figure 5. Average particle concentration data during the Arctic air mass period. a) Average
vertical profiles of $N_{5-20}$, $N_{20-40}$, $N_{>40}$, $N_{>80}$, and $N_{>150}$. b) Average (solid line) and median (dashed
line) size distribution within the BL and the FT. The light blue area represents the 25-75th %
percentile range. c) Frequency distribution of the occurrence of UFP illustrates the large variability
of the UFP concentrations within the BL.





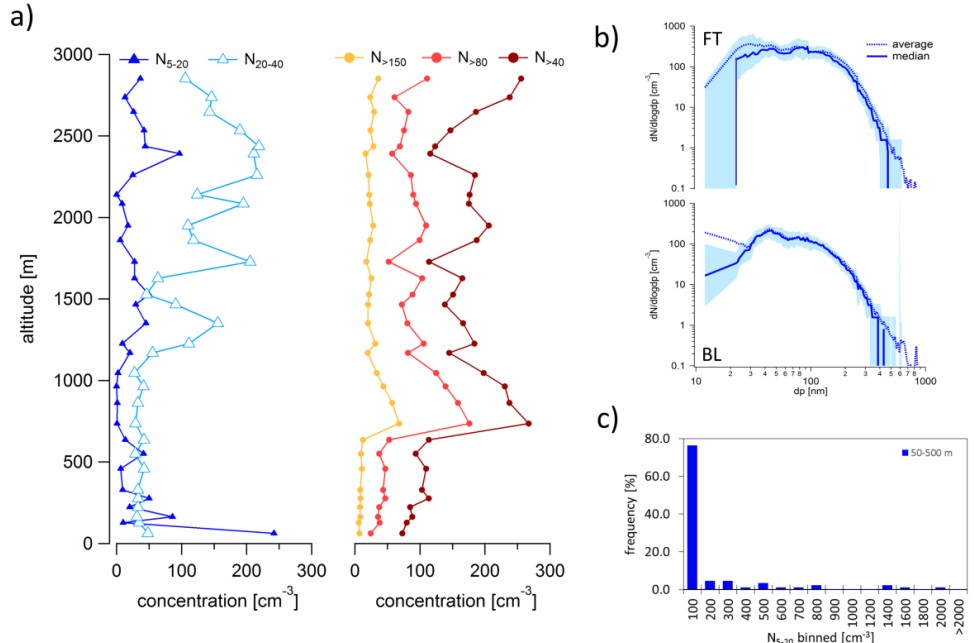


Figure 6. Average particle concentration data on the transition day. a) Average vertical profiles of $N_{5-20}$, $N_{20-40}$, $N_{>40}$, $N_{>80}$, and $N_{>150}$. b) Average (solid line) and median (dashed line) size distribution within the BL and the FT. The light blue area represents the 25-75th % percentile range. c) Frequency distribution of the occurrence of UFP illustrates the large variability of the UFP concentrations within the BL.






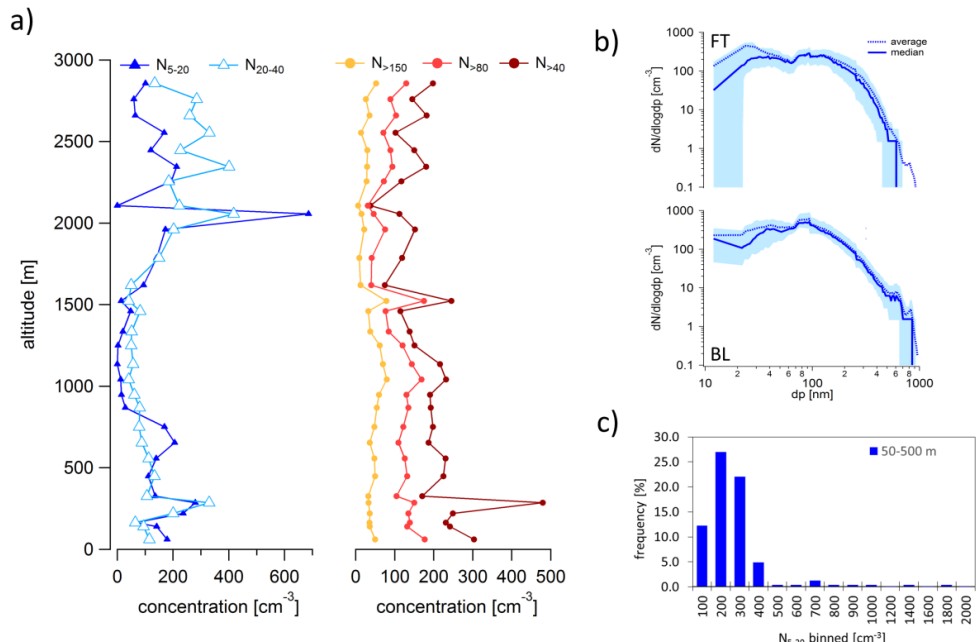


Figure 7. Average particle concentration data during the southern air mass period. a) Average
vertical profiles of $N_{5-20}$, $N_{20-40}$, $N_{>40}$, $N_{>80}$, and $N_{>150}$. b) Average (solid line) and median (dashed
line) size distribution within the BL and the FT. The light blue area represents the 25-75[th] %
percentile range. c) Frequency distribution of the occurrence of UFP illustrates the large variability
of the UFP concentrations within the BL.







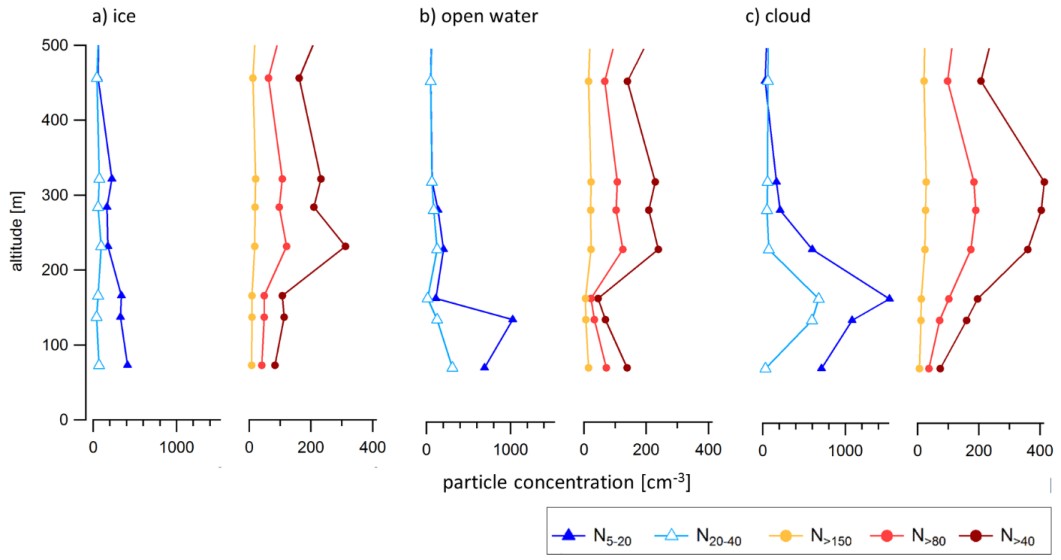


Figure 8. Average profiles of particle concentrations above ice, open water and cloud. The number
of data points for each specific profile is: 130 above water, 216 above cloud, and 123 above water.





Figure 9. Case study from July 8 flight. Time series of flight altitude and illustration of the surface
including cloud coverage (a), aerosol size (b) and chemical composition (c-e). (f) Temperature and
relative humidity profile near the locations i-vi (shown in a). (g) Wind direction and wind speed
for the entire period











Figure 10. (a) Time series of aircraft altitude color coded with the wind direction and time series of wind speed (b) FLEXPART-WRF seven day backwards potential emissions sensitivities for points along the flight track (60 second release at time at indicated time and location) showing the air mass history at 5 representative locations within the SIL. The plume centroid location for particles with age of one day is indicated. (c) The bottom plots show the altitude of plume centroid 48 hours back in time.





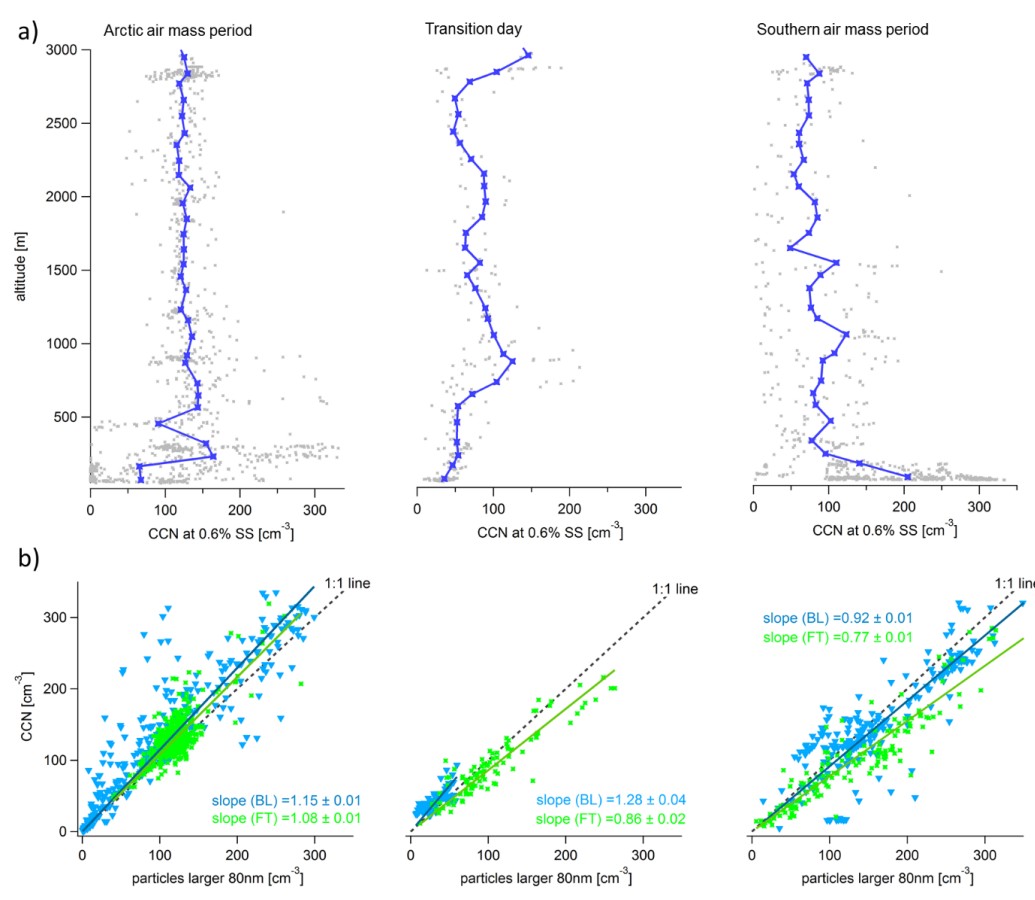


Figure 11. (a) Vertical profiles of average CCN concentrations (dark blue). All data points are
plotted in light grey. (b) Correlation plots between CCN concentrations and particles larger than
80nm.