# Peer review of "Summertime observations of elevated levels of ultrafine"

_Atmospheric Chemistry and Physics, 2016_

## Referee Comment (RC1)

General comments:

This is a very solid study with important new results on the aerosol in the summer Arctic and its possible sources and effects. Besides the few detailed comments listed below I only have one major issue. Why were the rich and unique data on aerosol composition only discussed in connection with the case study and not also in the discussions of air masses?

Recommendation: Accept with modifications and complementing discussion of aerosol composition.

| Line | Comment |
|------|---------|
| 46 | Please cite the references that established the phenomenon of Arctic haze long before your first citation |
| 52 | Again, the influence of the Arctic front has been published about decades before your first citation |
| 85 | Define "main mode number density" |
| 91 | There is another, more general, explanation, (see Heintzenberg and Leck, 1994) |
| 116 | To be complete you might want to cite Heintzenberg et al. (1991) even though no size distributions were measured |
| 151 | Where do these transmission data come from? |
| 152 | There must have been a substantial temperature increase from ambient to inside the CPC. Did you estimate the potential shrinking and loss of volatile particles due to this temperature increase? |
| 182 | "Floe"? |
| 384 | Explain "SIL" in text and figure |
| 545 | No "loadings", please! |
| 792 | No citations "in prep.", please! |

Literature

Heintzenberg, J., Ström, J., Ogren, J. A., and Fimpel, H.-P.: Vertical profiles of aerosol properties in the summer troposphere of the European Arctic, Atmos. Environ., 25A, 621-628, 1991.

Heintzenberg, J., and Leck, C.: Seasonal variation of the atmospheric aerosol near the top of the marine boundary layer over Spitsbergen related to the Arctic sulphur cycle, Tellus, 46B, 52-67, 1994.

---

## Referee Comment (RC2) · Anonymous Referee #2 · 27 Aug 2016

This paper investigates ultrafine particle (UFP) concentrations in the Canadian Arctic through aircraft measurements. The topic is of interest and important given the rapid changes expected for this region. Overall, I think an attempt at a more quantitative analysis, a more in-depth discussion of the limitations of the approach used in this work, (given that this has been grappled with and addressed in the past literature) and a more complete citation of the existing publications on this topic is needed.

A stated goal is to understand new particle formation (NPF) events (ie, the source of UFP) and how they evolve (grow) to affect radiative forcing through acting as CCN.

The paper really mainly focuses on the NPF events. A number or instruments were deployed, most important for addressing this question are measurements of all particle number concentrations larger than 5nm and number size distributions of particles larger than 20 nm. By difference, the authors determine the number between 5 and 20 nm to assess NPF. There is a substantial body of published literature on the the subject of investigating NFP using difference measurements, ie, typically this was done by subtracting data from a UFCPC (eg, TSI 3025) and CPC (eg, TSI 3010 or similar), to obtain concentrations of particle in the 3 to 10 nm range, and the limitations with this type of difference approach. In this study, these limitations are more severe as UFPs are defined over a larger size range, 5 to 20 nm, and smallest size is higher, 5 nm, further away from the size of the newly nucleated particle ($\sim$1 to 1.5 nm). Furthermore, there are no measurements of possible nucleation gas-phase precursors, such as sulfuric acid, (or SO2 and estimated OH could work, but not as well), ammonia and possible low volatile VOCs. Since new stable particles have sizes near 1 or so nm, and in pristine environments particle growth rates can be slow, it can take considerable time for the NPF event to be observable with the the deployed instrumentation. This make interpretation of this data very challenging for investigating nucleation conditions. Thus, essentially, the authors instrumentation is not ideal for addressing the scientific aim of defining NPF conditions in the Arctic.

First, it is recommended that the authors present a thorough assessment of the limitations with their method. This should include: 1) An analysis of the LOD of the difference method used to determine N(5-20) is needed. This will likely show that only significant concentrations of N(5-20) particles can be detected with certainty, potentially leading to a view biased by focusing only on larger nucleation events, which should be discussed. (This should likely include uncertainty in counting statistics, integrated size distributions to get total N of the measured distribution, N20, both instrument flow uncertainties, errors associated with subtracting two large numbers of small difference, etc. 2) The uncertainty in conditions that actually existed at time of the NPF event versus what existed at the time of the measurement due to expected times between the

two (ie, if possible, estimate the delay from nucleation to detection based on estimated or assumed growth rates of nanoparticles).

Without measurements of possible new particle precursor gases and lack of particle concentrations near the critical nucleation diameter it is very difficult to discuss conditions leading to NPF with certainty. For example, it is noted that often an inverse relation is seen between N(5-20) and larger particle concentrations. This could be due to formation of these smaller particles in cleaner air (ie, nucleation precursor vapors can build up due to lack of vapor condensational sink), or the new particles in the cleaner region just have a longer life span compared to particles that may have been formed in the other regions that had higher concentrations of pre-existing particles, which depleted the N(5-20) particles faster. Thus, how can one conclude that the only particle formation events are in the clean regions (possible confirmation bias).

I think the authors have provided evidence that UF particles are generated in the Arctic BL and it can occur under clean conditions, but what happens in more polluted air masses is not really known. Specifically, how (by what process), under what conditions (where), is not definitively known. More analysis, involving calculations (estimations) of lifetimes, more discussion of possible differences in measured total particle surface areas between regions, may allow more definitive conclusions. Some data is clearly not available to make more quantitative assessments, but maybe typical values reported in clean similar environments can set a bound.

Finally, it is not clear what is special about the clouds that leads to NPF in their vicinity. Since this is one major finding of this paper, more details are warranted, including a more comprehensive analysis of the literature, eg; [Clarke et al., 1999a; Clarke et al., 1999b; Hegg, 1991; Hegg et al., 1990; Hoppel et al., 1994; Mauldin et al., 1997; Perry and Hobbs, 1994; Radke and Hobbs, 1991; Raes, 1995; Weber et al., 2001; Wiedensohler et al., 1997]

Specifics.

Lines 73 to 76. These statements are unsupported and possibly not accurate. It all depends on the conditions (thus the statement must be qualified) and also depends on what is defined as ultrafine particles.

Should explicitly calculate estimated losses associated with sampling 5 nm particles (ie, this is the worst case scenario).

For given AMS LODs, what was the AMS sampling rate.

Was N(tot) every defined? Ie, what is the smallest size.

Line 339, what exactly does a burst of N(5-20) mean, was it a burst because the aircraft flow through a small region of high concentrations, or is it being implied that there was a burst in NPF? If the latter, how is that known? In fact, the use of "burst" throughout the paper is confusing since what it really means is isolated regions of high UF particles were encountered, not the typical meaning of a large nucleation event occurring over a short period of time (which I believe is what is implied). It is not clear that the latter can be determined from this sampling approach.

Lines 350. . . on possible reasons for high fractions of UF particles; this could also include, very slow growth rates, biased aircraft sampling (ie, you went looking for them).

Lines 462-465. This sentence seems to exactly confirm the limitations with the analysis used in the paper; the presence of UFP may have mostly to do with residence time in the BL and not mechanism that formed them.

Line 497-498; This conclusion is not the only possible one, it is also possible that they just are not scavenged and so have a longer life time (as noted already). Without measuring the nucleation precursor species, it is hard to tell. Calculations of life times of particles of 5 to 20nm due to coagulation, based on given size distributions may be insightful. Growth out of the 5 to 20 nm size range due to condensation is likely more important. Maybe if some assumptions are made on typical concentrations of H2SO4 or low volatile VOCS one could also estimate this, or use reported growth rates in other

clean regions?

Lines 573-575 and 594-595. Not clear what the all the ideal conditions for NPF were, is it just high OA levels? More details would be good, see references (eg, precipitating clouds scavenge accumulation mode aerosols, reducing surface area, but nucleation precursor gases can pass through (eg, SO2) forming H2SO4 in the high OH (high actinic flux and RH) in the cloud outflow, plus high RH leads to NPF. . ..

Lines 647-648: I think this sentence should be clarified. It cannot be said that the open water is the source of the observed particles, it maybe true, but it could also be that stable new particles of sub 5 nm size were formed elsewhere and just grew to your detectable size due to ocean emissions.

References:

Clarke, A. D., V. N. Kapustin, F. L. Eisele, R. J. Weber, and P. H. McMurry (1999a), Particle production near marine clouds: sulfuric acid and predictions from classical binary nucleation, Geophys. Res. Lett., 26, 2425-2428.

Clarke, A. D., F. Eisele, V. N. Kapustin, K. Moore, D. Tanner, L. Mauldin, M. Litchy, B. Lienert, M. A. Carroll, and G. Albercook (1999b), Nucleation in the equatorial free troposphere: Favorable environments during PEM-Tropics, J. Geophys. Res., 104, 5735-5744.

Hegg, D. A. (1991), Particle production in clouds, Geophys. Res. Let., 18, 995-998.

Hegg, D. A., L. F. Radke, and P. V. Hobbs (1990), Particle production associated with marine clouds, J. Geophys. Res., 95, 13917-13926.

Hoppel, W. A., G. M. Frick, J. W. Fitzgerald, and R. E. Larson (1994), Marine boundary layer measurements of new particle formation and the effects nonprecipitating clouds have on aerosol size distributions, J. Geophys. Res., 99, 14,443-414,459.

Mauldin, R. L., S. Madronich, S. J. Flocke, F. L. Eisele, G. J. Frost, and A. S. H. Prevot

(1997), New insights on OH: Measurements around and in clouds, Geophys. Res. Lett., 24(No 23), 3033-3036.

Perry, K. D., and P. V. Hobbs (1994), Further evidence for particle nucleation in clear air adjacent to marine cumulus clouds, J. Geophys. Res., 99, 22803-22818.

Radke, L. F., and P. V. Hobbs (1991), Humidity and particle fields around some small cumulus clouds, J. Atmos. Sci., 48, 1190-1193.

Raes, F. (1995), Entrainment of free tropospheric aerosols as a regulating mechanism for cloud condensation nuclei in the remote marine boundary layer, J. Geophys. Res., 100, 2893-2903.

Weber, R. J., G. Chen, D. D. Davis, R. L. Mauldin, D. J. Tanner, F. L. Eisele, A. D. Clarke, D. C. Thornton, and A. R. Bandy (2001), Measurements of enhanced H2SO4 and 3-4 nm particles near a frontal cloud during ACE 1, J. Geophys. Res., 106, 24,107-124,117.

Wiedensohler, A., H.-C. Hansson, D. Orsini, M. Wendisch, F. Wagner, K. N. Bower, T. W. Chourlarton, M. Wells, M. Parkin, K. Acker, W. Wieprecht, M. C. Facchini, J. A. Lind, S. Fuzzi, B. G. Arends, and M. Kulmala (1997), Night-time formation and occurrence of new particles associated with orographic clouds, Atmos. Environ., 31, 2545-2559.

---

## Referee Comment (RC3) · Anonymous Referee #3 · 5 Sep 2016

As a description of a flight campaign this manuscript includes a lot of information and the authors should have credit for trying to limit what is probably much more than enough behind the scene. Just as using cloud probes as a tool to stratify cloudy or non-cloudy measurement I see no problem of using differences between instruments as indicators for NPF (despite any measurement problems). The absolute numbers are really not followed up in the work. Hence, I will not dwell on measurement details. What I am missing is a Reader's Digest for modelers. Much of what is presented was already observed during previous campaigns, but the wealth of data could be presented in a summary nicely arranged with pertinent chemical and thermodynamically properties.

These cases could then be tried and tested using models. The aim in the beginning of the manuscript states a focus on UFP and this is ok, but quantifying their potential impact on impact requires a model. The processes are very complex, and any changes in cloud base height for instance will over compensate any aerosol effect. Again, a model is needed. I'm not convinced the CCN chapter of the manuscript is required for the NPF focus. In my opinion, the papers stands well as a description of the campaign, but I would prefer that the paper takes the understanding further than that of Shaw, Atmospheric Environment Vol. 23, No. 12, pp. 284-2846, 1989. What extra knowledge stands out form these flights besides, low mixing, low surface area, high insolation? A summary of this specifically would be a nice contribution. I don't contest that it is in the manuscript, but it could be summarized in a nice form. Details: Orography is a source for concern at Svalbard, what about the conditions at the flight campaign? Strom et al. 2009 fig 11 Tellus would be nice to compare directly with the supplement figure 1. The fact that Aitken mode particles are not observed right at the surface could be an instrument detection issue I guess. Particles must grow to detectable size. On the source of particle near the surface, have a look at: Lampert et al., Inclined Lidar Observations of Boundary Layer Aerosol Particles above the Kongsfjord, Svalbard As an example of ocean source. Acta Geophysica 60(5), October 2012

---

## Author Comment (AC1) · 15 Jan 2017

**Responses to the Reviewers**

We would like to thank all three reviewers for their thorough comments on this manuscript, which helped to improve the paper. Our responses to general and specific comments are below.

The comments of the reviewers are printed in bold. All line numbers in bold refer to the original manuscript, all others to the revised version.

**Reviewer 1**

Reviewer general comments: This is a very solid study with important new results on the aerosol in the summer Arctic and its possible sources and effects. Besides the few detailed comments listed below I only have one major issue. Why were the rich and unique data on aerosol composition only discussed in connection with the case study and not also in the discussions of air masses? Recommendation: Accept with modifications and complementing discussion of aerosol composition.

Response - We want to thank Dr. Heintzenberg for his comments on this manuscript. With respect to his major point, we agree that there is also a complementary data set from the AMS that flew on the POLAR6 during the summer campaign. As he points out, we include those data only in the discussion of the case study over the western end of Lancaster Sound. The major reason to not include the AMS dataset for the entire campaign is that the paper was largely focused on the ultrafine particles for which the AMS does not provide direct composition measurements (with the aerodynamic inlet measuring aerosol composition with unity transmission efficiency from just above 100 nm to 700 nm). As well, the AMS data set is extremely full and is being analyzed for presentation as either one or two separate, stand-alone papers. This future work will address the sulfate, organic, ammonium, and MSA aerosol mass concentrations in the campaign. As well, detailed sourcing via investigating the relationship of the aerosol composition to the time spent previously in marine boundary layers (or over other terrain) will be accomplished with FLEXPART-WRF analyses. This new analysis is well beyond what could have been included in the present paper, which is already quite long. We included just a small subset of the AMS data in the present paper to complete the discussion of the case study.

Each of the additional points is addressed below:

**Line 46 - Please cite the references that established the phenomenon of Arctic haze long before your first citation.**

Response – There are many such references, and we selected the following four for inclusion here on lines 49-50 of the revised manuscript:

- Rahn, K. A., Borys, R. D. and Shaw, G. E.: The Asian source of Arctic haze bands, Nature, 268, 713-715, doi: 10.1038/268713a0, 1977.
- Heintzenberg, B. J.: Particle size distribution and optical properties, Tellus, 32, 251–260, 10.1111/j.2153-3490.1980.tb00952.x, 1980.
- Shaw, G.E. and Stamnes, K.: Arctic haze: perturbations of the polar radiation budget. Ann. N. E Ad. Aci. 338, 533-539, doi: 10.1111/j.1749-6632.1980.tb17145.x 1980.

• Barrie, L. A.: Arctic air pollution: An overview of current knowledge, Atmos. Environ., 20(4), 643–663, doi:10.1016/0004-6981(86)90180-0, 1986.

**Line 52 - Again, the influence of the Arctic front has been published about decades before your first citation.**

Response – Here, reference to Barrie (1986) has been included on line 53.

**Line 85 - Define "main mode number density".**

Response – We have removed bracketed terms. The terms "accumulation" and "Aitken" are sufficient here.

**Line 91 - There is another, more general, explanation, (see Heintzenberg and Leck, 1994)**

Response – Reference added at line 96 with text as follows: "and marine biogenic sulphur (Heintzenberg and Leck, 1994)."

**Line 116 - To be complete you might want to cite Heintzenberg et al. (1991) even though no size distributions were measured.**

Response – Reference and the following text added on lines 122-126: "Although no size distribution measurements were performed, Heintzenberg et al. (1991) measured vertical profiles of the total particle number concentration greater than 10 nm during June and July, 1984 over the Fram Strait-Spitsbergen area, and found a "rather uniform distribution" with altitude. Their measurements, however, were confined to 500 m-MSL and above."

**Line 151 - Where do these transmission data come from?**

Response – Reference to Leaitch et al. (2016) added on line 162.

**Line 152 - There must have been a substantial temperature increase from ambient to inside the CPC. Did you estimate the potential shrinking and loss of volatile particles due to this temperature increase?**

Response – We have not, but we have added a brief discussion of this point beginning on line 162: "Although the transfer of the aerosol from outside to the instruments is relatively fast (5 seconds and less), volatilization of some components of the particles may have occurred. However, it has been demonstrated that the growth of smaller particles by organic condensation occurs primarily by low volatility organic components (e.g. Pierce et al., 2012). Thus, the integrity of the smaller particles that are the focus of the discussion here is more likely to have been maintained. We do expect increasing line losses of particles with sizes decreasing from 10 nm. Therefore our observations will underestimate  $N_{5-20}$ ."

**Line 182 - "Floe"?**

Response - Corrected to "flow" on line 198.

**Line 384 - Explain "SIL" in text and figure**

Response - SIL removed, and replaced with defined BL (boundary layer).

**Line 545 - No "loadings", please!**

Response - Changed 'loadings' to "mass concentrations" on lines 577-578.

Line 792 - No citations "in prep.", please! Response – Removed from reference list and changed to personal communication on line 590-591 and on line 710.

---

## Author Comment (AC3) · 15 Jan 2017

Responses to the Reviewers

We would like to thank all three reviewers for their thorough comments on this manuscript, which helped to improve the paper. Our responses to general and specific comments are below.

The comments of the reviewers are printed in bold. All line numbers in bold refer to the original manuscript, all others to the revised version.

Reviewer 3

We thank the reviewer for their comments.

**Reviewer: As a description of a flight campaign this manuscript includes a lot of information and the authors should have credit for trying to limit what is probably much more than enough behind the scene. Just as using cloud probes as a tool to stratify cloudy or non-cloudy measurement I see no problem of using differences between instruments as indicators for NPF (despite any measurement problems). The absolute numbers are really not followed up in the work. Hence, I will not dwell on measurement details.**

Response: We thank the reviewer for their perspectives on the measurement approaches used in the study. We also believe also that there is merit to using the difference between the numbers of particles between 5 and 20 nm, i.e. $N_{5-20}$, to study the nature of aerosol processes as a function of location and height in this high Arctic regime. In our response to Reviewer 2, we discuss further the merits of this approach.

**Reviewer: What I am missing is a Reader's Digest for modelers. Much of what is presented was already observed during previous campaigns, but the wealth of data could be presented in a summary nicely arranged with pertinent chemical and thermodynamically properties.**

Response: This is a very good point, and so we have re-written the abstract of the paper and tightened the language in the conclusions section. Overall, we believe that the main points ("Readers Digest") of the study that a modeler should take away are that: 1) new particle formation occurs readily in the Canadian high Arctic boundary layer, a region dominated by marine and coastal regions, 2) particle growth also occurs in these regions under specific environments, 3) the highest levels of ultrafine particles were associated with above-cloud conditions influenced by marine air, and 4) ultrafine particle formation occurs much less frequently in the free troposphere under these conditions. Modeling efforts would ideally represent such behavior but are currently limited by our knowledge of marine aerosol precursor emissions.

That all said, we actually disagree that much of what we have observed has been seen before. In particular, this is the first systematic altitude-resolved study of the nature of ultrafine particles in mid-summer in the high Arctic. As we responded to Reviewer 2, what is important here is that relative to the cloud-free observations over ice and water, the UFP over cloud are common and the associated concentrations are higher (Figure 8). Also, in both the cloud and open water cases, the highest UFP concentrations are found at the lowest measurement levels, implying that the

surface (water or cloud) is critical to the NPF process. It is information of this type that is needed for comparisons against model output, to test the validity of the model representations of aerosol processes.

**Reviewer: These cases could then be tried and tested using models. The aim in the beginning of the manuscript states a focus on UFP and this is ok, but quantifying their potential impact requires a model. The processes are very complex, and any changes in cloud base height for instance will over compensate any aerosol effect. Again, a model is needed.**

Response: While the impacts of the UPFs are certainly interesting and have motivated this study to a large degree, it was beyond the scope of this observational paper to include the impacts that can only be evaluated with a model. However, we make reference to the work of Leaitch et al. (2016) that has pointed out that particles as small as 20 nm become activated into cloud droplets in this environment, motivating potential impacts and the needed to understand the processes that lead to their formation. Also, we now refer to Croft et al. (2016) that models one significant impact of NPF on Arctic radiative forcing.

**Reviewer: I'm not convinced the CCN chapter of the manuscript is required for the NPF focus. In my opinion, the papers stands well as a description of the campaign, but I would prefer that the paper takes the understanding further than that of Shaw, Atmospheric Environment Vol. 23, No. 12, pp. 284-2846, 1989. What extra knowledge stands out form these flights besides, low mixing, low surface area, high insolation? A summary of this specifically would be a nice contribution. I don't contest that it is in the manuscript, but it could be summarized in a nice form.**

Response: We agree that Shaw nicely illustrated that particle nucleation may occur in clean atmospheric environments, such as those in polar regions that have experienced recent scavenging. So, in that context, we fully agree that there are no new conceptual findings in our work compared to this work by Shaw, and other, earlier studies. However, what is new in this work are actual measurements of ultrafine particles in the high Arctic summer, especially in an altitude-resolved manner. These have not been documented so clearly before and such information is needed to compare against model output. Further, the work contrasts NPF over three different surfaces (ice, water and top of low cloud) in the same environment, which has never been done before and is important. We don't have sufficient statistics to conclude that the low cloud presence enhances the NPF relative to open water, but that certainly is the indication.

While we also agree that the CCN measurements are in some sense disconnected from the focus on the paper on the UFPs, we prefer to leave them in the paper as an illustration of the numbers of particles that may be arising, in part, from the growth of the UFPs that were measured. In order to improve the connection between the UFP observations and the CCN, we have added Figure 12 that shows correlations among the smaller particle sizes and the CCN. We also emphasise that the CCN measured here are larger than the average size of particle found by Leaitch et al. (2016) to participate in cloud droplet nucleation.

**Reviewer: Orography is a source for concern at Svalbard, what about the conditions at the flight campaign?**

Response: This is an interesting point re. orography. It is true that the nucleation and growth event documented in Willis et al. occurred in air that had resided over Devon Island (maximum altitude 2000 m) before descending through katabatic flow to the Lancaster Sound, and the same is evident in the event documented in Figure 8 of the present paper. Such air may have been cleaned by passing through this higher elevation location, lowering its condensation sink. However, aside from whatever reduced the condensation sink, the surfaces appear to be the sources of the particle precursors.

**Reviewer: Ström et al. 2009 fig 11 Tellus would be nice to compare directly with the supplement figure 1. The fact that Aitken mode particles are not observed right at the surface could be an instrument detection issue I guess. Particles must grow to detectable size. On the source of particle near the surface, have a look at: Lampert et al., Inclined Lidar Observations of Boundary Layer Aerosol Particles above the Kongsfjord, Svalbard As an example of ocean source. Acta Geophysica 60(5), October 2012.**

Response: This is a very valuable point, that it is possible that particles nucleate at the surface but require time to growth to sizes that are detectable. However, as referred to in our response to Reviewer 2, we apologize because we have now improved our profile averaging approach. We have revised Figure 8, which originally showed that the maximum in the UFP concentration over open water was above the lowest sampling level. That was due in part to a bias associated with the averaging time of the SMS, which has been removed. To Figure 8, we have now added $N_{tot}$, which was sampled every second and shows the increase in particles over open water is also associated with the lowest sampling level. That observation is consistent with the results of Willis et al. (2016) as well as past observations related to polynyas (Leaitch et al. 1983 and 1994).

References

Leaitch, W.R., Hoff, R.M., Melnichuk, S., and Hogan, W.: Some chemical and physical properties of the Arctic winter aerosol in northeastern Canada. J. Climate Appl. Meteorol., 23, 916-928, http://dx.doi.org/10.1175/1520-0450(1984)023<0916:SPACPO>2.0.CO;2, 1984.

Leaitch, W.R., Barrie, L.A., Bottenheim, J.W., Li, S.-M., Shepson, P. and Yokouchi, Y.: Airborne observations related ozone depletion at polar sunrise. J. Geophys. Res., 99, 25499-25517, 10.1029/94JD02750, 1994.

Willis, M. D., Burkart, J., Thomas, J. L., Köllner, F., Schneider, J., Bozem, H., Hoor, P. M., Aliabadi, A. A., Schulz, H., Herber, A. B., Leaitch, W. R. and Abbatt, J. P. D.: Growth of nucleation mode particles in the summertime Arctic: a case study, Atmos. Chem. Phys., 7663–7679, doi:10.5194/acp-16-7663-2016, 2016.

---

## Author Comment (AC4) · 16 Jan 2017

Revised Manuscript - January 15[th], 2017

Changes are highlighted in yellow.

[revised manuscript text omitted]

---

## Author Comment (AC2)

**Responses to the Reviewers**

We would like to thank all three reviewers for their thorough comments on this manuscript, which helped to improve the paper. Our responses to general and specific comments are below.

The comments of the reviewers are printed in bold. All line numbers in bold refer to the original manuscript, all others to the revised version.

**Reviewer 2**

Reviewer general comments: This paper investigates ultrafine particle (UFP) concentrations in the Canadian Arctic through aircraft measurements. The topic is of interest and important given the rapid changes expected for this region. Overall, I think an attempt at a more quantitative analysis, a more in-depth discussion of the limitations of the approach used in this work, (given that this has been grappled with and addressed in the past literature) and a more complete citation of the existing publications on this topic is needed.

A stated goal is to understand new particle formation (NPF) events (ie, the source of UFP) and how they evolve (grow) to affect radiative forcing through acting as CCN.

The paper really mainly focuses on the NPF events. A number or instruments were deployed, most important for addressing this question are measurements of all particle number concentrations larger than 5nm and number size distributions of particles larger than 20 nm. By difference, the authors determine the number between 5 and 20 nm to assess NPF. There is a substantial body of published literature on the the subject of investigating NFP using difference measurements, ie, typically this was done by subtracting data from a UFCPC (eg, TSI 3025) and CPC (eg, TSI 3010 or similar), to obtain concentrations of particle in the 3 to 10 nm range, and the limitations with this type of difference approach. In this study, these limitations are more severe as UFPs are defined over a larger size range, 5 to 20 nm, and smallest size is higher, 5 nm, further away from the size of the newly nucleated particle (1 to 1.5 nm). Furthermore, there are no measurements of possible nucleation gas-phase precursors, such as sulfuric acid, (or SO2 and estimated OH could work, but not as well), ammonia and possible low volatile VOCs. Since new stable particles have sizes near 1 or so nm, and in pristine environments particle growth rates can be slow, it can take considerable time for the NPF event to be observable with the the deployed instrumentation. This make interpretation of this data very challenging for investigating nucleation conditions. Thus, essentially, the authors instrumentation is not ideal for addressing the scientific aim of defining NPF conditions in the Arctic.

First, it is recommended that the authors present a thorough assessment of the limitations with their method. This should include: 1) An analysis of the LOD of the difference method used to determine N(5-20) is needed. This will likely show that only significant concentrations of N(5-20) particles can be detected with certainty, potentially leading to a view biased by focusing only on larger nucleation events, which should be discussed. (This should likely include uncertainty in counting statistics, integrated size distributions to get total N of the measured distribution, N20, both instrument flow uncertainties, errors associated with subtracting two large numbers of small difference, etc. 2) The uncertainty in conditions that actually existed at time of the NPF event versus what

existed at the time of the measurement due to expected times between the two (ie, if possible, estimate the delay from nucleation to detection based on estimated or assumed growth rates of nanoparticles).

Without measurements of possible new particle precursor gases and lack of particle concentrations near the critical nucleation diameter it is very difficult to discuss conditions leading to NPF with certainty. For example, it is noted that often an inverse relation is seen between N(5-20) and larger particle concentrations. This could be due to formation of these smaller particles in cleaner air (ie, nucleation precursor vapors can build up due to lack of vapor condensational sink), or the new particles in the cleaner region just have a longer life span compared to particles that may have been formed in the other regions that had higher concentrations of pre-existing particles, which depleted the N(5-20) particles faster. Thus, how can one conclude that the only particle formation events are in the clean regions (possible confirmation bias). I think the authors have provided evidence that UF particles are generated in the Arctic BL and it can occur under clean conditions, but what happens in more polluted air masses is not really known. Specifically, how (by what process), under what conditions (where), is not definitively known. More analysis, involving calculations (estimations) of lifetimes, more discussion of possible differences in measured total particle surface areas between regions, may allow more definitive conclusions. Some data is clearly not available to make more quantitative assessments, but maybe typical values reported in clean similar environments can set a bound.

Finally, it is not clear what is special about the clouds that leads to NPF in their vicinity. Since this is one major finding of this paper, more details are warranted, including a more comprehensive analysis of the literature, eg; [Clarke et al., 1999a; Clarke et al., 1999b; Hegg, 1991; Hegg et al., 1990; Hoppel et al., 1994; Mauldin et al., 1997; Perry and Hobbs, 1994; Radke and Hobbs, 1991; Raes, 1995; Weber et al., 2001; Wiedensohler et al., 1997]

Responses – We thank the reviewer for their comments, which help to improve the paper. The following are a few responses to the reviewer's more general comments:

1) As discussed below, we have included some of the additional references the reviewer suggests.

2) We agree with the reviewer that growth rates may be slower in pristine environments. However, in this study we were apparently very close to a source of particle precursors (biologically productive sea and sea-ice interface). In other words, we were not necessarily pristine from the point of view of potential precursors, even though we were pristine by comparison with anthropogenic environments. The work of Willis et al. (2016) illustrates rapid aerosol growth in this environment.

The main point of the paper is to demonstrate that ultrafine particles are associated with the ocean surface during the Arctic summer when particle mass concentrations are extremely low. Airborne measurements of NPF associated with open water in the Arctic are surprisingly few: Leaitch et al. (1983 and 1994). This work offers a unique perspective on the issue. With respect to growth of the particles to reach 5 nm and beyond 20 nm, we note that in the absence of significant coagulation (the effect of which will be reduced due to the relatively low

concentrations of particles smaller than 20 nm) it requires more than 100 times more condensable precursor for particles to grow from 20 nm to 50 nm compared with particles growing from 1 nm to 10nm. Willis et al. (2016) showed that for this environment particles were able to grow to 50 nm low over the open water over approximately one hour, which means that the growth from 1 nm to 10 nm can occur over one minute or less, justifying our use of instruments that are sensitive to 5 nm particles and larger. Of course, we would have very much have liked to have instruments operating sensitive to 1 nm particles but they were not available to us for this study. It is our belief that doing the first systematic altitude-resolved study of particle size distributions in the summertime Arctic marine environment with a standard set of instrumentation makes this work noteworthy.

Also, please note that we may have misled readers (and the reviewer) of the ACPD version of the paper by averaging all data in our vertical profiles, which led to a bias by overweighting single flights at some altitude intervals. We apologize for this. The new averaging method used in the revised manuscript equally weights data from all flights at all altitudes intervals. We added a brief description to the paper (line 244-249): "An average profile for a single flight was obtained by binning all data from the respective flight into altitude intervals of 100m starting at the lowest flight altitude. In addition to data obtained during vertical profile flights, data acquired while flying at a constant level were also included. Average profiles containing data of more than one flight were calculated by averaging the respective single flight profiles. "We revised all figures showing average profiles (Figure 5-8).

Figure 5 and Figure 8 originally showed that the maximum in the UFP concentration over open water was above the lowest sampling level. To the new version of Figure 8, we have added  $N_{tot}$ , which was sampled every second and shows the increase in particles over open water is also associated with the lowest sampling level. That observation is consistent with the results of Willis et al. (2016) as well as past observations related to polynyas (Leaitch et al. 1983 and 1994).

As for the limit of detection, Leaitch et al. (2013) found an average CPC-SMPS of 47/cc for points during dark periods at Alert, Nunavut (1861 one hour averages) with a standard deviation of 73/cc and a 95th percentile of 144/cc, indicating that values of CPC-SMPS in excess of 200/cc are reasonable indicators for NPF. Our discussion on lines 338-345 of the original manuscript (363-370 of revised paper) already addresses this concern.

3) With respect to the reviewer's comments concerning the conditions under which we observe NPF, we now indicate in the Abstract that high levels of UFPs are observed when the condensation sink is smallest, i.e. lowest numbers of larger particles. We feel that this is an accurate short summary, based on the observations made in the study. The reviewer suggests that the "precursor vapors can build up due to the lack of a condensation sink" but it is not clear what sort of 'build up' is needed. The formation of 1000 particles/cc, each of which is 10 nm in diameter, requires very little precursor. Over a biologically productive sea, it is very likely that sufficient precursor will be available for NPF if the condensation sink is low enough, as in the early part of this study.

The reviewer suggests that "the new particles in the cleaner region just have a longer life span compared to particles that may have been formed in the other regions that had higher concentrations of pre-existing particles, which depleted the N(5-20) particles faster." If that were the case, then we would have seen the NPF occur at other locations and not principally associated with the ocean surface.

Finally, the reviewer argues that "what happens in more polluted air masses is not really known". This may be true but we find less evidence for NPF associated with less pristine conditions. It is clear that in this relatively clean background environment, a low condensation sink is a necessary condition. The paper does not address more polluted conditions.

4) As in our response to the reviewer's comment concerning lines 573-575, we have added a sentence to elaborate on the potential role of clouds in this instance (lines 608-611 of the revised manuscript). We use two of the references suggested by the reviewer (one reference the reviewer suggested was already present): Clarke et al. (1999) and Mauldin et al. (1997). These are on lines 601-602 of the revised manuscript. Of course it is not unique that NPF is associated with clouds. We also added two more references suggested by the reviewer that illustrate a connection of new particles with clouds (Radke and Hobbs 1991, Wiedensohler et al. 1997). What is important here is that relative to the cloud-free observations over ice and water, the UFP over cloud were common and the associated concentrations are higher (Figure 8).

References added:

- Clarke, A. D., Kapustin, V. N., Eisele, F. L., Weber, R. J., and McMurry, P. H.: Particle production near marine clouds: sulfuric acid and predictions from classical binary nucleation, Geophys. Res. Lett., 26, 2425-2428, doi: 10.1029/1999GL900438, 1999.
- Mauldin, R. L., Madronich, S., Flocke, S. J., Eisele, F. L., Frost, G. J. and Prevot, A. S. H. : New insights on OH: Measurements around and in clouds, Geophys. Res. Lett., 24(No 23), 3033-3036, doi:10.1029/97GL02983, 1997.
- Radke, F. L. and Hobbs, P. V.: Humidity and particle fields around some small cumulus clouds, Journal of atmospheric sciences, 48(9), 1190-1193, doi: http://dx.doi.org/10.1175/1520-0469(1991)048<1190:HAPFAS>2.0.CO;2, 1991.
- Wiedensohler, A. H.-C. Hansson, D. Orsini, M. Wendisch, F. Wagner, K.N. Bower, T.W. Chourlarton, M. Wells, M. Parkin, K. Acker, W. Wieprecht, M.C. Facchini, J.A. Lind, S. Fuzzi, B.G. Arends, M. Kulmalao: Night-time formation and occurrence of new particles associated with orographic clouds, Atmos. Env., 31(16), 2445-2559, doi: <a href="http://dx.doi.org/10.1016/S1352-2310(96)00299-3">http://dx.doi.org/10.1016/S1352-2310(96)00299-3</a>, 1997.

Specifics:

Reviewer - Lines 73 to 76. These statements are unsupported and possibly not accurate. It all depends on the conditions (thus the statement must be qualified) and also depends on what is defined as ultrafine particles.

Response – We feel these statements are supported by the observations, as discussed above.

**Reviewer - Should explicitly calculate estimated losses associated with sampling 5 nm particles (ie, this is the worst case scenario).**

Response – In response to Reviewer 1 and this comment, we have added on lines 159-165 of the revised paper: "Although the transfer of the aerosol from outside to the instruments is relatively fast (5 seconds and less), volatilization of some components of the particles may have occurred. However, it has been demonstrated that the growth of smaller particles by organic condensation occurs primarily by low volatility organic components (e.g. Riipinen et al., 2011; Pierce et al., 2012). Thus, the integrity of the smaller particles that are the focus of the discussion here is more likely to have been maintained. We do expect increasing line losses of particles with sizes decreasing from 10 nm. Therefore our observations will underestimate  $N_{5-20}$ ."

**Reviewer - For given AMS LODs, what was the AMS sampling rate.**

Response – The sampling rate was 30 seconds. This has now been added to the paper (line 222).

**Reviewer - Was N(tot) every defined? Ie, what is the smallest size.**

Response – Yes, it was defined on line 179 of the ACPD manuscript as the UCPC measurement, which on line 176 refers to particles larger than 5 nm. In addition, we have added on lines 194-196: ", noting as above that diffusional losses of particles smaller than 10 nm make the Ntot observations lower limits."

Reviewer - Line 339, what exactly does a burst of N(5-20) mean, was it a burst because the aircraft flow through a small region of high concentrations, or is it being implied that there was a burst in NPF? If the latter, how is that known? In fact, the use of "burst" throughout the paper is confusing since what it really means is isolated regions of high UF particles were encountered, not the typical meaning of a large nucleation event occurring over a short period of time (which I believe is what is implied). It is not clear that the latter can be determined from this sampling approach.

Response – Bursts here refers to sudden increases in N5-20. We have clarified that on line 359-361 of the revised manuscript by the addition of "Here we refer to "bursts" of particles as a sudden and relatively large increase in  $N_{5-20}$ . This may reflect inhomogeneities in the NPF process or may reflect the aircraft flying in and out of a NPF event."

**Reviewer - Lines 350: on possible reasons for high fractions of UF particles; this could also include, very slow growth rates, biased aircraft sampling (ie, you went looking for them).**

Response – As above, the results in Willis et al. (ACP, 2016) do not suggest very slow growth rates, and we do not include that as a possibility. We agree that there could be some bias associated with our sampling, and accordingly we have added the following sentence to lines 379-381 of the revised manuscript: "Also, since observations of UFP were one focus of this study, the fractional occurrence of the UFP mode may be slightly biased due to longer sampling times associated with UFP occurrence."

Reviewer - Lines 462-465. This sentence seems to exactly confirm the limitations with the analysis used in the paper; the presence of UFP may have mostly to do with residence time in the BL and not mechanism that formed them.

Response – If these particles were resident for a long time in the BL, then we should have observed them everywhere. But this was not consistent with the observations.

Reviewer - Line 497-498; This conclusion is not the only possible one, it is also possible that they just are not scavenged and so have a longer life time (as noted already). Without measuring the nucleation precursor species, it is hard to tell. Calculations of life times of particles of 5 to 20nm due to coagulation, based on given size distributions may be insightful. Growth out of the 5 to 20 nm size range due to condensation is likely more important. Maybe if some assumptions are made on typical concentrations of H2SO4 or low volatile VOCS one could also estimate this, or use reported growth rates in other clean regions?

Response – Our response is the same as to the comment immediately above as well as our above responses to the reviewer's general comments.

Reviewer - Lines 573-575 and 594-595. Not clear what the all the ideal conditions for NPF were, is it just high OA levels? More details would be good, see references (eg, precipitating clouds scavenge accumulation mode aerosols, reducing surface area, but nucleation precursor gases can pass through (eg, SO2) forming H2SO4 in the high OH (high actinic flux and RH) in the cloud outflow, plus high RH leads to NPF...

Response – Thank you. We have added the following on lines 608-611 of the revised manuscript: " In other words, precipitating clouds scavenge aerosol particles, reducing the surface area for condensation, but some fraction of nucleation precursor gases with lower Henry's Law constants can pass through (e.g.  $SO_2$ ) leaving the potential for  $H_2SO_4$  in the higher OH in the cloud outflow (a discussion of the processes can be found in Seinfeld and Pandis, 1998)."

Reviewer - Lines 647-648: I think this sentence should be clarified. It cannot be said that the open water is the source of the observed particles, it may be true, but it could also be that stable new particles of sub 5 nm size were formed elsewhere and just grew to your detectable size due to ocean emissions.

Response – See response to above comment concerning lines 462-465 and our above responses to your general comments.

**References**

Leaitch, W.R., Hoff, R.M., Melnichuk, S., and Hogan, W.: Some chemical and physical properties of the Arctic winter aerosol in northeastern Canada. J. Climate Appl. Meteorol., 23, 916-928, http://dx.doi.org/10.1175/1520-0450(1984)023<0916:SPACPO>2.0.CO;2, 1984.

Leaitch, W.R., Barrie, L.A., Bottenheim, J.W., Li, S.-M., Shepson, P. and Yokouchi, Y.: Airborne observations related ozone depletion at polar sunrise. J. Geophys. Res., 99, 25499-25517, 10.1029/94JD02750, 1994.

Leaitch, W. R., Sharma, S., Huang, L., Toom-Sauntry, D., Chivulescu, A., Macdonald, A. M., von Salzen, K., Pierce, J. R., Bertram, A. K., Schroder, J. C., Shantz, N. C., Chang, R. Y. W. and Norman, A.-L.: Dimethyl sulfide control of the clean summertime Arctic aerosol and cloud, Elem. Sci. Anth., 1(1), 17, doi:10.12952/journal.elementa.000017, 2013.

Willis, M. D., Burkart, J., Thomas, J. L., Köllner, F., Schneider, J., Bozem, H., Hoor, P. M., Aliabadi, A. A., Schulz, H., Herber, A. B., Leaitch, W. R. and Abbatt, J. P. D.: Growth of nucleation mode particles in the summertime Arctic: a case study, Atmos. Chem. Phys., 7663–7679, doi:10.5194/acp-16-7663-2016, 2016.